# Learning Markov State Abstractions
# for Deep Reinforcement Learning

**Cameron Allen**[*]
Brown University

**Neev Parikh**
Brown University

**Omer Gottesman**
Brown University

**George Konidaris**
Brown University

## Abstract

A fundamental assumption of reinforcement learning in Markov decision processes (MDPs) is that the relevant decision process is, in fact, Markov. However, when MDPs have rich observations, agents typically learn by way of an abstract state representation, and such representations are not guaranteed to preserve the Markov property. We introduce a novel set of conditions and prove that they are sufficient for learning a Markov abstract state representation. We then describe a practical training procedure that combines inverse model estimation and temporal contrastive learning to learn an abstraction that approximately satisfies these conditions. Our novel training objective is compatible with both online and offline training: it does not require a reward signal, but agents can capitalize on reward information when available. We empirically evaluate our approach on a visual gridworld domain and a set of continuous control benchmarks. Our approach learns representations that capture the underlying structure of the domain and lead to improved sample efficiency over state-of-the-art deep reinforcement learning with visual features—often matching or exceeding the performance achieved with hand-designed compact state information.

## 1 Introduction

Reinforcement learning (RL) in Markov decision processes with rich observations requires a suitable state representation. Typically, such representations are learned implicitly as a byproduct of doing deep RL. However, in domains where precise and succinct expert state information is available, agents trained on such expert state features usually outperform agents trained on rich observations. Much recent work (Shelhamer et al., 2016; Pathak et al., 2017; Ha & Schmidhuber, 2018; Gelada et al., 2019; Yarats et al., 2019; Kaiser et al., 2020; Laskin et al., 2020a,b; Zhang et al., 2021) has sought to close this *representation gap* by incorporating a wide range of representation-learning objectives that help the agent learn abstract representations with various desirable properties.

Perhaps the most obvious property to incentivize in a state representation is the Markov property, which holds if and only if the representation contains enough information to accurately characterize the rewards and transition dynamics of the decision process. Markov decision processes (MDPs) have this property by definition, and most reinforcement learning algorithms depend on having Markov state representations. For instance, the ubiquitous objective of learning a stationary, state-dependent optimal policy that specifies how the agent should behave is *only* appropriate for *Markov* states.

But learned abstract state representations are not necessarily Markov, even when built on top of MDPs. This is due to the fact that abstraction necessarily throws away information. Discard too much information, and the resulting representation cannot accurately characterize the environment. Discard too little, and agents will fail to close the representation gap. Abstraction must balance between

---

[*]Please send any correspondence to Cameron Allen <csal@brown.edu>. Code repository available at https://github.com/camall3n/markov-state-abstractions.

35th Conference on Neural Information Processing Systems (NeurIPS 2021).

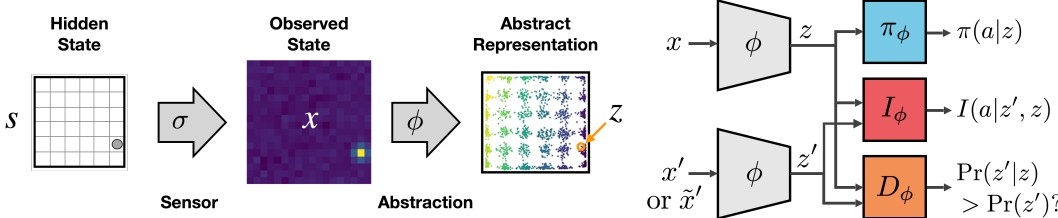

Figure 1: (Left) A $6 \times 6$ visual gridworld domain with hidden state $s$ and unknown sensor $\sigma$, where an abstraction function $\phi$ maps each high-dimensional observed state $x$ to a lower-dimensional abstract state $z$ (orange circle). (Right) Our Markov abstraction training architecture. A shared encoder $\phi$ maps ground states $x, x'$ to abstract states $z, z'$, which are inputs to an inverse dynamics model $I$ and a contrastive model $D$ that discriminates between real and fake state transitions. The agent's policy $\pi$ depends only on the current abstract state.

ignoring irrelevant information and preserving what is important for decision making. If reward feedback is available, an agent can use it to determine which state information is relevant to the task at hand. Alternatively, if the agent can predict raw environment observations from learned abstract states, then *all* available information is preserved (along with the Markov property). However, these approaches are impractical when rewards are sparse or non-existent, or observations are sufficiently complex.

We introduce a new approach to learning Markov state abstractions. We begin by defining a set of theoretical conditions that are sufficient for an abstraction to retain the Markov property. We next show that these conditions are approximately satisfied by simultaneously training an inverse model to predict the action distribution that explains two consecutive states, and a discriminator to determine whether two given states were in fact consecutive. Our combined training objective (architecture shown in Fig. 1, right) supports learning Markov abstract representations without requiring reward information or observation prediction.

Our method is effective for learning Markov state abstractions that are highly beneficial for decision making. We perform evaluations in two settings with rich (visual) observations: a gridworld navigation task (Fig. 1, left) and a set of continuous control benchmarks. In the gridworld, we construct an abstract representation offline—without access to reward feedback—that captures the underlying structure of the domain and fully closes the representation gap between visual and expert features. In the control benchmarks, we combine our training objective (online) with traditional RL, where it leads to a significant performance improvement over state-of-the-art visual representation learning.

## 2 Background

A Markov decision process $M$ consists of sets of states $X$ and actions $A$, reward function $R : X \times A \times X \rightarrow \mathbb{R}$, transition dynamics $T : X \times A \rightarrow \Pr(X)$, and discount factor $\gamma$. For our theoretical results, we assume $M$ is a Block MDP (Du et al., 2019) whose behavior is governed by a much smaller (but unobserved) set of states $S$, and where $X$ is a rich observation generated by a noisy sensor function $\sigma : S \rightarrow \Pr(X)$, as in Figure 1 (left). Block MDPs conveniently avoid potential issues arising from partial observability by assuming that each observation uniquely identifies the unobserved state that generated it. In other words, there exists a perfect "inverse sensor" function $\sigma^{-1}(x) \mapsto s$, which means the observations are themselves Markov, as we define below. Note that $S$, $\sigma$, and $\sigma^{-1}$ are all unknown to the agent.

**Definition 1 (Markov State Representation).** *A decision process $M = (X, A, R, T, \gamma)$ and its state representation $X$ are* Markov *if and only if $T^{(k)} \left( x_{t+1} | \{a_{t-i}, x_{t-i}\}_{i=0}^{k} \right) = T(x_{t+1}|a_t, x_t)$ and $R^{(k)} \left( x_{t+1}, \{a_{t-i}, x_{t-i}\}_{i=0}^{k} \right) = R(x_{t+1}, a_t, x_t)$, for all $a \in A$, $x \in X$, $k \geq 1$.*

The superscript $(k)$ denotes that the function is being conditioned on $k$ additional steps of history.

The Markov property means that each state $X$ it is a sufficient statistic for predicting the next state and expected reward, for any action the agent might select. Technically, each state must also be sufficient for determining the set of actions available to the agent in that state, but here we assume, as is common, that every action is available in every state.

The behavior of an RL agent is typically determined by a (Markov) policy $\pi : X \to \Pr(A)$, and each policy induces value function $V^\pi : X \to \mathbb{R}$, which is defined as the expected sum of future discounted rewards starting from a given state and following the policy $\pi$ thereafter. The agent's objective is to learn an optimal policy $\pi^*$ that maximizes value at every state. Note that the assumption that the optimal policy is stationary and Markov—that it only depends on state—is appropriate only if the decision process itself is Markov; almost all RL algorithms simply assume this to be true.

## 2.1 State Abstraction

To support decision making when $X$ is too high-dimensional for tractable learning, we turn to state abstraction. Our objective is to find an abstraction function $\phi : X \to Z$ mapping each ground state[2] $x$ to an abstract state $z = \phi(x)$, with the hope that learning is tractable using the abstract representation $Z$ (see Figure 1, left). Since our goal is to support effective abstract decision making, we are mainly concerned with the policy class $\Pi_\phi$, the set of policies with the same behavior for all ground states that share the same abstract state:

$$\Pi_\phi := \left\{ \pi : \big(\phi(x_1) = \phi(x_2)\big) \implies \big(\pi(a|x_1) = \pi(a|x_2)\big), \ \forall \, a \in A; x_1, x_2 \in X \right\}. \tag{1}$$

An abstraction $\phi : X \to Z$, when applied to an MDP $M$, induces a new abstract decision process $M_\phi = (Z, A, T^\pi_{\phi,t}, R^\pi_{\phi,t}, \gamma)$, whose dynamics may depend on the current timestep $t$ or the agent's behavior policy $\pi$, and, crucially, which is not necessarily Markov. Consider the following example.

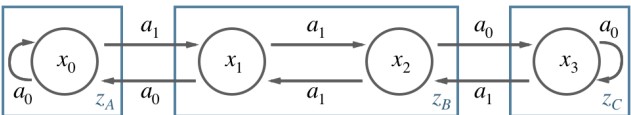

Figure 2: An MDP and a non-Markov abstraction.

The figure above depicts a four-state, two-action MDP, and an abstraction $\phi$ where $\phi(x_1) = \phi(x_2) = z_B$. It is common to view state abstraction as aggregating or partitioning ground states into abstract states in this way (Li et al., 2006). Generally this involves choosing a fixed weighting scheme $w(x)$ to express how much each ground state $x$ contributes to its abstract state $z$, where the weights sum to 1 for the set of $x$ in each $z$. We can then define the abstract transition dynamics $T_\phi$ as a $w$-weighted sum of the ground dynamics (and similarly for reward): $T_\phi(z'|a, z) = \sum_{x' \in z'} \sum_{x \in z} T(x'|a, x)w(x)$. A natural choice for $w(x)$ is to use the ground-state visitation frequencies. For example, if the agent selects actions uniformly at random, this leads to $w(x_0) = w(x_3) = 1$ and $w(x_1) = w(x_2) = 0.5$.

In this formulation, the abstract decision process is assumed to be Markov by construction, and $T_\phi$ and $R_\phi$ are assumed not to depend on the policy or timestep. But this is an oversimplification. The abstract transition dynamics are *not* Markov; they change depending on how much history is conditioned on: $T_\phi(z_A|a_t = a_0, z_t = z_B) = 0.5$, whereas $\Pr(z_A|a_t = a_0, z_t = z_B, a_{t-1} = a_1, z_{t-1} = z_A) = 1$. By contrast, the ground MDP's dynamics are fully deterministic (and Markov). Clearly, if we define the abstract MDP in this way, it may not match the behavior of the original MDP.[3] Even worse, if the agent's policy changes—such as during learning—this discrepancy can cause RL algorithms with bounded sample complexity in the ground MDP to make an arbitrarily large number of mistakes in the abstract MDP (Abel et al., 2018).

For the abstract decision process to faithfully simulate the ground MDP's dynamics, $w(x)$ must be allowed to vary such that it always reflects the correct ground-state frequencies. Unfortunately, even in the simple example above, maintaining accurate weights for $w(x)$ requires keeping track of an unbounded amount of history: if an agent reaches abstract state $z_B$ and repeats action $a_1$ an arbitrarily large number of times $(N)$, then knowing precisely which ground state it will end up in $(x_1$ or $x_2)$ requires remembering the abstract state it originally came from $N + 1$ steps prior. Successful modeling of the transition dynamics for a subsequent action $a_0$ hinges on exactly this distinction between $x_1$ and $x_2$. The abstraction introduces partial observability, and to compensate, $w(x)$ must be replaced with a belief distribution over ground states, conditioned on the entire history; instead

---

[2]We refer to $x \in X$ as *ground states* (and $M$ as the *ground MDP*), to reflect that these quantities are *grounded*, as opposed to abstract, i.e. they have a firm basis in the true environment.

[3]Abel et al. (2018) presented a three-state chain MDP where they made a similar observation.

of an abstract MDP, we have an abstract POMDP (Bai et al., 2016). This is especially unsatisfying, not just because learning is challenging in POMDPs (Zhang et al., 2012), but because formulating the problem as a Block MDP was supposed to avoid precisely this type of partial observability: an abstraction exists (namely $\phi = \sigma^{-1}$) that would result in a fully observable abstract MDP, if only the agent knew what it was.

We show that it is possible to learn a state abstraction that both reflects the behavior of the underlying ground MDP *and* preserves the Markov property in the abstract MDP, without ever estimating or maintaining a belief distribution. Our theoretical result leverages inverse dynamics models and contrastive learning to derive conditions under which a state abstraction is Markov. We then adapt these conditions into a corresponding training objective for learning such an abstraction directly from the agent's experience in the ground MDP.

## 3  Related Work

### 3.1  Bisimulation

The idea of learning Markov state abstractions is related to the concept of bisimulation (Dean & Givan, 1997), the strictest type of state aggregation discussed in Li et al. (2006), where ground states are equivalent if they have exactly the same expected reward and transition dynamics. Preserving the Markov property is a prerequisite for a bisimulation abstraction, since the abstraction must also preserve the (Markov) ground-state transition dynamics. Bisimulation-based abstraction is appealing because, by definition, it leads to high-fidelity representations. But bisimulation is also very restrictive, because it requires $\phi$ to be Markov for *any* policy (rather than just those in $\Pi_\phi$).

Subsequent work on approximate MDP homomorphisms (Ravindran & Barto, 2004) and bisimulation metrics (Ferns et al., 2004, 2011) relaxed these strict assumptions and allowed ground states to have varying degrees of "bisimilarity." Castro (2020) introduced a further relaxation, $\pi$-bisimulation, which measures the behavioral similarity of states *under a policy* $\pi$. But whereas full bisimulation can be too strong, since it constrains the representation based on policies the agent may never actually select, $\pi$-bisimulation can be too weak, since if the policy deviates from $\pi$ (e.g. during learning), the metric must be updated, and the representation along with it. Our approach can be thought of as a useful compromise between these two extremes.

While bisimulation-based approaches have historically been computationally expensive and difficult to scale, recent work has started to change that (Castro, 2020; Lehnert & Littman, 2020; Van der Pol et al., 2020; Biza et al., 2021). Two recent algorithms in particular, DeepMDP (Gelada et al., 2019) and Deep Bisimulation for Control (DBC) (Zhang et al., 2021), learn approximate bisimulation abstractions by training the abstraction end-to-end with an abstract transition model and reward function. This is a rather straightforward way to learn Markov abstract state representations since it effectively encodes Definition 1 as a loss function.

One drawback of bisimulation-based methods is that learning an accurate model can be challenging and typically requires restrictive modeling assumptions, such as deterministic, linear, or Gaussian transition dynamics. Bisimulation methods may also struggle if rewards are sparse or if the abstraction must be learned without access to rewards. Jointly training an abstraction $\phi$ with only the transition model $\widehat{T}(\phi(x), a) \approx \phi(x')$ can easily lead to a trivial abstraction like $\phi(x) \mapsto 0$ for all $x$, since $\phi$ produces both the inputs and outputs for the model. Our approach to learning a Markov abstraction avoids this type of representation collapse without learning a forward model, and is less restrictive than bisimulation, since it is compatible with both reward-free and reward-informed settings.

### 3.2  Ground-State Prediction and Reconstruction

Ground-state (or pixel) prediction (Watter et al., 2015; Song et al., 2016; Kaiser et al., 2020) mitigates representation collapse by forcing the abstract state to be sufficient not just for predicting future *abstract* states, but also future *ground states*. Unfortunately, in stochastic domains, this comes with the challenging task of density estimation over the ground state space, and as a result, performance is about on-par with end-to-end deep RL (Van Hasselt et al., 2019). Moreover, both pixel prediction and the related task of pixel reconstruction (Mattner et al., 2012; Finn et al., 2016; Higgins et al., 2017; Corneil et al., 2018; Ha & Schmidhuber, 2018; Yarats et al., 2019; Hafner et al., 2020; Lee et al.,

2020) are misaligned with the fundamental goal of state abstraction. These approaches train models to perfectly reproduce the relevant ground state, ergo the abstract state must effectively throw away no information. By contrast, the objective of state abstraction is to throw away *as much information as possible*, while preserving only what is necessary for decision making. Provided the abstraction is Markov and accurately simulates the ground MDP, we can safely discard the rest of the observation.

### 3.3 Inverse Dynamics Models

As an alternative to (or in addition to) learning a forward model, it is sometimes beneficial to learn an inverse model. An inverse dynamics model $I(a|x', x)$ predicts the distribution over actions that could have resulted in a transition between a given pair of states. Inverse models have been used for improving generalization from simulation to real-world problems (Christiano et al., 2016), enabling effective robot motion planning (Agrawal et al., 2016), defining intrinsic reward bonuses for exploration (Pathak et al., 2017; Choi et al., 2019), and decoupling representation learning from rewards (Zhang et al., 2018). But while inverse models often help with representation learning, we show in Sec. 4.2 that they are insufficient for ensuring a Markov abstraction.

### 3.4 Contrastive Learning

Since the main barrier to effective next-state prediction is learning an accurate forward model, a compelling alternative is contrastive learning (Gutmann & Hyvärinen, 2010), which sidesteps the prediction problem and instead simply aims to decide whether a particular state, or sequence of states, came from one distribution or another. Contrastive loss objectives typically aim to distinguish either sequential states from non-sequential ones (Shelhamer et al., 2016; Anand et al., 2019; Stooke et al., 2020), real states from predicted ones (Van den Oord et al., 2018), or determine whether two augmented views came from the same or different observations (Laskin et al., 2020b). Contrastive methods learn representations that in some cases lead to empirically substantial improvements in learning performance, but none has explicitly addressed the question of whether the resulting state abstractions actually preserve the Markov property. We are the first to show that without forward model estimation, pixel prediction/reconstruction, or dependence on reward, the specific combination of inverse model estimation and contrastive learning that we introduce in Section 4 is sufficient to learn a Markov abstraction.

### 3.5 Kinematic Inseparability

One contrastive approach which turns out to be closely related to Markov abstraction is Misra et al.'s (2020) HOMER algorithm, and the corresponding notion of *kinematic inseparability* (KI) abstractions. Two states $x'_1$ and $x'_2$ are defined to be kinematically inseparable if $\Pr(x, a|x'_1) = \Pr(x, a|x'_2)$ and $T(x''|a, x'_1) = T(x''|a, x'_2)$ (which the authors call "backwards" and "forwards" KI, respectively). The idea behind KI abstractions is that unless two states can be distinguished from each other—by either their backward or forward dynamics—they ought to be treated as the same abstract state. The KI conditions are slightly stronger than the ones we describe in Section 4, although when we convert our conditions into a training objective in Section 5, we additionally satisfy a novel form of the KI conditions, which helps to prevent representation collapse. While our approach works for both continuous and discrete state spaces, HOMER was only designed for discrete abstract states, and requires specifying—in advance—an upper bound on the *number* of abstract states (which is impossible for continuous state spaces), as well as learning a "policy cover" to reach each of those abstract states (which remains impractical even under discretization).

(For a more detailed discussion about KI and Markov abstractions, see Appendix F.)

### 3.6 Other Approaches

The Markov property is just one of many potentially desirable properties that a representation might have. Not all Markov representations are equally beneficial for learning; otherwise, simply training an RL agent end-to-end on (frame-stacked) image inputs ought to be sufficient, and none of the methods in this section would need to do representation learning at all.

Smoothness is another desirable property and its benefits in RL are well known (Pazis & Parr, 2013; Pirotta et al., 2015; Asadi et al., 2018). Both DeepMDP (Gelada et al., 2019) and DBC (Zhang

et al., 2021), which we compare against, utilize Lipschitz smoothness when learning abstract state representations. We find in Section 7 that a simple smoothness objective helps our approach in a similar way. A full investigation of other representation-learning properties (e.g. value preservation (Abel et al., 2016), symbol construction (Konidaris et al., 2018), suitability for planning (Kurutach et al., 2018), information compression (Abel et al., 2019)) is beyond the scope of this paper.

Since our approach does not require any reward information and is agnostic as to the underlying RL algorithm, it would naturally complement exploration methods designed for sparse reward problems (Pathak et al., 2017; Burda et al., 2018). Exploration helps to ensure that the experiences used to learn the abstraction cover as much of the ground MDP's state space as possible. In algorithms like HOMER (above) and the more recent Proto-RL (Yarats et al., 2021), the exploration and representation learning objectives are intertwined, whereas our approach is, in principle, compatible with any exploration algorithm. Here we focus solely on the problem of learning Markov state abstractions and view exploration as an exciting direction for future work.

## 4  Markov State Abstractions

Recall that for a state representation to be Markov (whether ground or abstract), it must be a sufficient statistic for predicting the next state and expected reward, for any action the agent selects. The state representation of the ground MDP is Markov by definition, but learned state abstractions typically have no such guarantees. In this section, we introduce conditions that provide the missing guarantees.

Accurate abstract modeling the ground MDP requires replacing the fixed weighting scheme $w(x)$ of Section 2 with a belief distribution, denoted by $B_\phi(x|\{\cdots\})$, that measures the probability of each ground state $x$, conditioned on the entire history of agent experiences. Our objective is to find an abstraction $\phi$ such that any amount of history can be summarized with a single abstract state $z$.

When limited to the most recent abstract state $z$, $B_\phi$ may be policy-dependent and non-stationary:[4]

$$B_{\phi,t}^\pi(x|z) := \frac{\mathbb{1}[\phi(x) = z]\, P_t^\pi(x)}{\sum_{\tilde{x} \in z} P_t^\pi(\tilde{x})}, \qquad P_t^\pi(x) := \sum_{a \in A} \sum_{\tilde{x} \in X} T(x|a, \tilde{x}) \pi_{t-1}(a|\tilde{x}) P_{t-1}^\pi(\tilde{x}), \quad (2)$$

for $t \geq 1$, where $\mathbb{1}[\cdot]$ denotes the indicator function, $\pi$ is the agent's (possibly non-stationary) behavior policy, and $P_0$ is an arbitrary initial state distribution. Note that $P_t^\pi$ and $B_{\phi,t}^\pi$ may still be non-stationary even if $\pi$ is stationary.[5]

We generalize to a $k$-step belief distribution (for $k \geq 1$) by conditioning (2) on additional history:

$$B_{\phi,t}^{\pi(k)} \left(x_t|z_t, \{a_{t-i}, z_{t-i}\}_{i=1}^k\right) :=$$
$$\frac{\mathbb{1}[\phi(x_t) = z_t] \sum_{x_{t-1} \in X} T(x_t|a_{t-1}, x_{t-1}) B_{\phi,t}^{\pi(k-1)} \left(x_{t-1} \mid z_{t-1}, \{a_{t-i}, z_{t-i}\}_{i=2}^k\right)}{\sum_{\tilde{x}_t \in z_t} \sum_{\tilde{x}_{t-1} \in z_{t-1}} T(\tilde{x}_t|a_{t-1}, \tilde{x}_{t-1}) B_{\phi,t}^{\pi(k-1)} \left(\tilde{x}_{t-1} \mid z_{t-1}, \{a_{t-i}, z_{t-i}\}_{i=2}^k\right)}, \quad (3)$$

where $B_{\phi,t}^{\pi(0)} := B_{\phi,t}^\pi$. Any abstraction induces a belief distribution, but the latter is only independent of history for a *Markov* abstraction. We formalize this concept with the following definition.

**Definition 2 (Markov State Abstraction).** *Given an MDP $M = (X, A, R, T, \gamma)$, initial state distribution $P_0$, and policy class $\Pi_C$, a state abstraction $\phi : X \to Z$ is* Markov *if and only if for any policy $\pi \in \Pi_C$, $\phi$ induces a belief distribution $B_\phi^\pi$ such that for all $x \in X$, $z \in Z$, $a \in A$, and $k \geq 1$:*
$$B_{\phi,t}^{\pi(k)} \left(x|z_t, \{a_{t-i}, z_{t-i}\}_{i=1}^k\right) = B_{\phi,t}^\pi \left(x|z_t\right).$$

In other words, Markov abstractions induce belief distributions that only depend on the most recent abstract state. This property allows an agent to avoid belief distributions entirely, and base its decisions solely on abstract states. Note that Definition 2 is stricter than Markov state representations

---

[4]This section closely follows Hutter (2016), except here we consider belief distributions over ground states, rather than full histories. An advantage of Hutter's work is that it also considers *abstractions* over histories, though it only provides a rough sketch of how to learn such abstractions. Extending our learning objective to support histories is a natural direction for future work.

[5]This can happen, for example, when the policy induces either a Markov chain that does not have a stationary distribution, or one whose stationary distribution is different from $P_0$.

(Def. 1). An abstraction that collapses every ground state to a single abstract state still produces a Markov state representation, but for non-trivial ground MDPs it also induces a history-dependent belief distribution.

Given these definitions, we can define the abstract transitions and rewards for the policy class $\Pi_\phi$ (see Eqn. (1)) as follows:[6]

$$T_{\phi,t}^\pi(z'|a,z) = \sum_{x'\in z'} \sum_{x\in z} T(x'|a,x)B_{\phi,t}^\pi(x|z), \tag{4}$$

$$R_{\phi,t}^\pi(z',a,z) = \sum_{x'\in z'} \sum_{x\in z} \frac{R(x',a,x)T(x'|a,x)}{T_{\phi,t}^\pi(z'|a,z)} B_{\phi,t}^\pi(x|z). \tag{5}$$

Conditioning the belief distribution on additional history yields $k$-step versions compatible with Definition 1. In the special case where $B_\phi(x|z)$ is stationary and policy-independent (and if rewards are defined over state-action pairs), we recover the fixed weighting function $w(x)$ of Li et al. (2006).

## 4.1 Sufficient Conditions for a Markov Abstraction

The strictly necessary conditions for ensuring an abstraction $\phi$ is Markov over its policy class $\Pi_\phi$ depend on $T$ and $R$, which are typically unknown and hard to estimate due to $X$'s high-dimensionality. However, we can still find sufficient conditions without explicitly knowing $T$ and $R$. To do this, we require that two quantities are equivalent in $M$ and $M_\phi$: the inverse dynamics model, and a density ratio that we define below. The inverse dynamics model $I_t^\pi(a|x',x)$ is defined in terms of the transition function $T(x'|a,x)$ and expected next-state dynamics $P_t^\pi(x'|x)$ via Bayes' theorem: $I_t^\pi(a|x',x) := \frac{T(x'|a,x)\pi_t(a|x)}{P_t^\pi(x'|x)}$, where $P_t^\pi(x'|x) = \sum_{\tilde{a}\in A} T(x'|\tilde{a},x)\pi_t(\tilde{a}|x)$. The same is true of their abstract counterparts, $I_{\phi,t}^\pi(a|z',z)$ and $P_{\phi,t}^\pi(z'|z)$.

**Theorem 1.** *If $\phi : X \to Z$ is a state abstraction of MDP $M = (X, A, R, T, \gamma)$ such that for any policy $\pi$ in the policy class $\Pi_\phi$, the following conditions hold for every timestep $t$:*

1. ***Inverse Model.*** *The ground and abstract inverse models are equal: $I_{\phi,t}^\pi(a|z',z) = I_t^\pi(a|x',x)$, for all $a \in A$; $z, z' \in Z$; $x, x' \in X$, such that $\phi(x) = z$ and $\phi(x') = z'$.*

2. ***Density Ratio.*** *The ground and abstract next-state density ratios are equal, when conditioned on the same abstract state: $\frac{P_t^\pi(z'|z)}{P_t^\pi(z')} = \frac{P_t^\pi(x'|z)}{P_t^\pi(x')}$, for all $z, z' \in Z$; $x' \in X$, such that $\phi(x') = z'$, where $P_t^\pi(x'|z) = \sum_{\tilde{x}\in X} P_t^\pi(x'|\tilde{x})B_{\phi,t}^\pi(\tilde{x}|z)$, and $P_t^\pi(z') = \sum_{\tilde{x}'\in X} P_t^\pi(\tilde{x}')B_{\phi,t}^\pi(\tilde{x}'|z')$.*

*Then $\phi$ is a Markov state abstraction.*

**Corollary 1.1.** *If $\phi : X \to Z$ is a Markov state abstraction of MDP $M = (X, A, R, T, \gamma)$ over the policy class $\Pi_\phi$, then the abstract decision process $M_\phi = (Z, A, R_{\phi,t}^\pi, T_{\phi,t}^\pi, \gamma)$ is also Markov.*

We defer all proofs to Appendix D.

Theorem 1 describes a pair of conditions under which $\phi$ is a Markov abstraction. Of course, the conditions themselves do not constitute a training objective—we can only use them to confirm an abstraction is Markov. In Section 5, we adapt these conditions into a practical representation learning objective that is differentiable and suitable for learning $\phi$ using deep neural networks. First, we show why the Inverse Model condition alone is insufficient.

## 4.2 An Inverse Model Counterexample

The example MDP in Figure 2 additionally demonstrates why the Inverse Model condition alone is insufficient to produce a Markov abstraction. Observe that any valid transition between two ground states uniquely identifies the selected action. The same is true for abstract states since the only way to

---

[6]For the more general definitions that support arbitrary policies, see Appendix C.

reach $z_B$ is via action $a_1$, and the only way to leave is action $a_0$. Therefore, the abstraction satisfies the Inverse Model condition for any policy. However, as noted in Section 2.1, conditioning on additional history changes the abstract transition probabilities, and thus the Inverse Model condition is not sufficient for an abstraction to be Markov. In fact, we show in Appendix D.2 that, given the Inverse Model condition, the Density Ratio condition is actually *necessary* for a Markov abstraction.

## 5    Training a Markov Abstraction

We now present a set of training objectives for approximately satisfying the conditions of Theorem 1. Since the theorem applies for the policy class $\Pi_\phi$ induced by the abstraction, we restrict the policy by defining $\pi$ as a mapping from $Z \to \Pr(A)$, rather than from $X \to \Pr(A)$. In cases where $\pi$ is defined implicitly via the value function, we ensure that the latter is defined over abstract states.

**Inverse Models.**    To ensure the ground and abstract inverse models are equal, we consider a batch of $N$ experiences $(x_i, a_i, x_i')$, encode ground states with $\phi$, and jointly train a model $f(a|\phi(x_i'), \phi(x_i); \theta_f)$ to predict a distribution over actions, with $a_i$ as the label. This can be achieved by minimizing a cross-entropy loss, for either discrete or continuous action spaces:

$$\mathcal{L}_{Inv} := -\frac{1}{N} \sum_{i=1}^{N} \log f(a = a_i | \phi(x_i'), \phi(x_i); \theta_f).$$

Note that because the policy class is restricted to $\Pi_\phi$, if the policy is stationary and deterministic, then $I_{\phi,t}^\pi(a|z', z) = \pi_\phi(a|z) = \pi(a|x) = I_t^\pi(a|x', x)$ and the Inverse Model condition is satisfied trivially. Thus we expect $\mathcal{L}_{Inv}$ to be most useful for representation learning when the policy has high entropy or is changing rapidly, such as during early training.

**Density Ratios.**    The second condition, namely that $\frac{P_{\phi,t}^\pi(z'|z)}{P_{\phi,t}^\pi(z')} = \frac{P_t^\pi(x'|z)}{P_t^\pi(x')}$, means we can distinguish conditional samples from marginal samples equally well for abstract states or ground states. This objective naturally lends itself to a type of contrastive loss. We generate a batch of $N$ sequential state pairs $(x_i, x_i')$ as samples of $\Pr(x'|x)$, and a batch of $N$ non-sequential state pairs $(x_i, \tilde{x}_i')$ as samples of $\Pr(x')$, where the latter pairs can be obtained, for example, by shuffling the $x_i'$ states in the first batch. We assign positive labels $(y_i = 1)$ to sequential pairs and negative labels to non-sequential pairs. This setup, following the derivation of Tiao (2017), allows us to write density ratios in terms of class-posterior probabilities: $\delta(x') := \frac{\Pr(x'|x)}{\Pr(x')} = \frac{p(y=1|x,x')}{1-p(y=1|x,x')}$ and $\delta_\phi(z') := \frac{\Pr(z'|z)}{\Pr(z')} = \frac{q(y=1|z,z')}{1-q(y=1|z,z')}$, where $p$ and $q$ are just names for specific probability distributions.[7] We jointly train an abstraction $\phi$ and a classifier $g(y|\phi(x'), \phi(x); \theta_g)$, minimizing the cross-entropy between predictions and labels $y_i$:

$$\mathcal{L}_{Ratio} := -\frac{1}{2N} \sum_{i=1}^{2N} \log g(y = y_i | \phi(x_i'), \phi(x_i); \theta_g).$$

In doing so, we ensure $g$ approaches $p$ and $q$ simultaneously, which drives $\delta_\phi(z') \to \delta(x')$.

Note that this is stronger than the original condition, which only required the ratios to be equal in expectation. This stronger objective, when combined with the inverse loss, actually encodes a novel form of the kinematic inseparability conditions from Section 3.5, which further helps to avoid representation collapse. (See Appendix F for more details.)

**Smoothness.**    To improve robustness and encourage our method to learn smooth representations like those discussed in Section 3.6, we optionally add an additional term to our loss function:

$$\mathcal{L}_{Smooth} := (\text{ReLU}(\|\phi(x') - \phi(x)\|_2 - d_0))^2.$$

This term penalizes consecutive abstract states for being more than some predefined distance $d_0$ away from each other. Appendix L describes an additional experiment and provides further justification for why representation smoothness is an important consideration that complements the Markov property.

---

[7]For completeness, we reproduce the derivation in Appendix E.

**Markov Abstraction Objective.**  We generate a batch of experiences using a mixture of abstract policies $\pi_i \in \Pi_C \subseteq \Pi_\phi$ (for example, with a uniform random policy), then train $\phi$ end-to-end while minimizing a weighted combination of the inverse, ratio, and smoothness losses:

$$\mathcal{L}_{Markov} := \alpha \mathcal{L}_{Inv} + \beta \mathcal{L}_{Ratio} + \eta \mathcal{L}_{Smooth},$$

where $\alpha$, $\beta$, and $\eta$ are coefficients that compensate for the relative difficulty of minimizing each individual objective for the domain in question.

The Markov objective avoids the problem of representation collapse without requiring reward information or ground state prediction. A trivial abstraction like $\phi(x) \mapsto 0$ would not minimize $\mathcal{L}_{Markov}$, because it contains no useful information for predicting actions or distinguishing authentic transitions from manufactured ones.

## 6  Offline Abstraction Learning for Visual Gridworlds

First, we evaluate our approach for learning an abstraction offline for a visual gridworld domain (Fig. 1, left). Each discrete $(x, y)$ position in the $6 \times 6$ gridworld is mapped to a noisy image (see Appendix G). We emphasize that the agent only sees these images; it does not have access to the ground-truth $(x, y)$ position. The agent gathers a batch of experiences in a version of the gridworld with *no rewards or terminal states*, using a uniform random exploration policy over the four directional actions.

These experiences are then used offline to train an abstraction function $\phi_{Markov}$, by minimizing $\mathcal{L}_{Markov}$ (with $\alpha = \beta = 1$, $\eta = 0$). We visualize the learned 2-D abstract state space in Figure 3a (top row) and compare against ablations that train with only $\mathcal{L}_{Inv}$ or $\mathcal{L}_{Ratio}$, as well as against two baselines that we train via pixel prediction and reconstruction, respectively (see Appendix I for more visualizations). We observe that $\phi_{Markov}$ and $\phi_{Inv}$ cluster the noisy observations and recover the $6 \times 6$ grid structure, whereas the others do not generally have an obvious interpretation. We also observed that $\phi_{Ratio}$ and $\phi_{Autoenc}$ frequently failed to converge.

Next we froze these abstraction functions and used them to map images to abstract states while training DQN (Mnih et al., 2015) on the resulting features. We measured the learning performance of each pre-trained abstraction, as well as that of end-to-end DQN with no pretraining. We plot learning curves in Figure 3b. For reference, we also include learning curves for a uniform random policy and DQN trained on ground-truth $(x, y)$ position with no abstraction.

Markov abstractions match the performance of ground-truth position, and beat every other learned representation except $\phi_{Inv}$. Note that while $\phi_{Markov}$ and $\phi_{Inv}$ perform similarly in this domain, there is no reason to expect $L_{Inv}$ to work on its own for other domains, since it lacks the theoretical motivation of our combined Markov loss. When the combined loss is minimized, the Markov conditions are satisfied. But *even if* the inverse loss goes to zero on its own, the counterexample in Section 4.2 demonstrates that this is insufficient to learn a Markov abstraction.

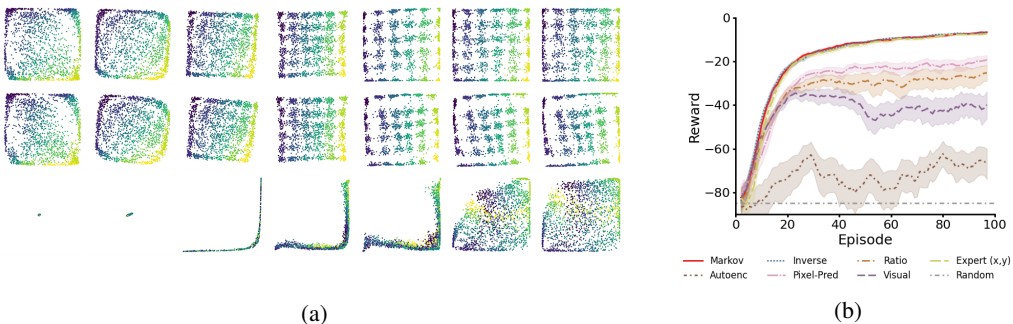

(a)                                                                                  (b)

Figure 3: (a) Visualization of learning progress at selected times (left to right) of a 2-D state abstraction for the $6 \times 6$ visual gridworld domain: (top row) $\mathcal{L}_{Markov}$; (middle row) $\mathcal{L}_{Inv}$ only; (bottom row) $\mathcal{L}_{Ratio}$ only. Color denotes ground-truth $(x, y)$ position, which is not shown to the agent. (b) Mean episode reward for the visual gridworld navigation task. Markov abstractions significantly outperform end-to-end training with visual inputs, and match the performance of the expert $(x, y)$ position features. (300 seeds; 5-point moving average; shaded regions denote 95% confidence intervals.)

# 7   Online Abstraction Learning for Continuous Control

Next, we evaluate our approach in an online setting with a collection of image-based, continuous control tasks from the DeepMind Control Suite (Tassa et al., 2020). Our training objective is agnostic about the underlying RL algorithm, so we use as our baseline the state-of-the-art technique that combines Soft Actor-Critic (SAC) (Haarnoja et al., 2018) with random data augmentation (RAD) (Laskin et al., 2020a). We initialize a replay buffer with experiences from a uniform random policy, as is typical, but before training with RL, we use those same experiences *with reward information removed* to pretrain a Markov abstraction. We then continue training with the Markov objective alongside traditional RL. (See Appendix H for implementation details).

In Figure 4, we compare against unmodified RAD, as well as contrastive methods CURL (Laskin et al., 2020b) and CPC (Van den Oord et al., 2018), bisimulation methods DeepMDP (Gelada et al., 2019) and DBC (Zhang et al., 2021), and pixel-reconstruction method SAC-AE (Yarats et al., 2019). As a reference, we also include non-visual SAC with expert features. All methods use the same number of environment steps (the experiences used for pretraining are not additional experiences).

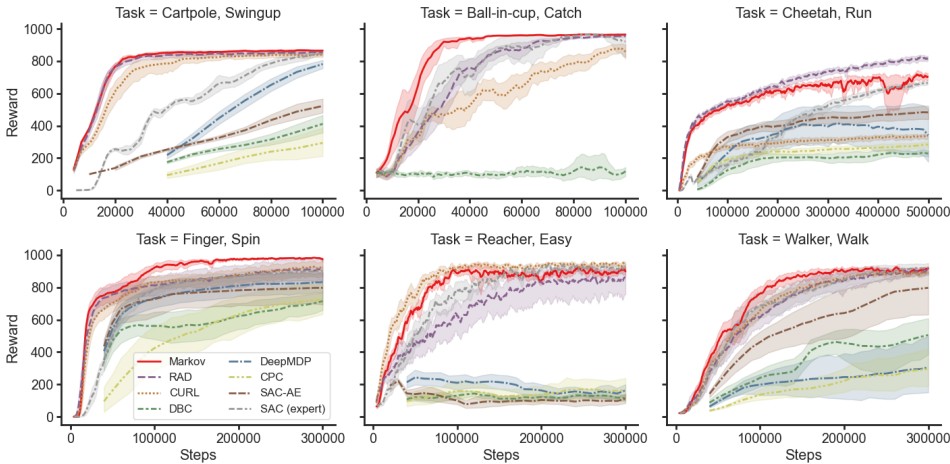

Figure 4: Mean episode reward vs. environment steps for DeepMind Control. Adding our Markov objective leads to improved learning performance. (10 seeds; 5-point moving average; shaded regions denote 90% confidence intervals; learning curve data is available at the linked code repository.)

Relative to RAD, our method learns faster on four domains and slower on one, typically achieving the same final performance (better in one, worse in one). It performs even more favorably relative to the other baselines, of which CURL is most similar to our method, since it combines contrastive learning with data augmentation similar to that of RAD.[8] Our approach even represents a marginal improvement over a hypothetical "best of" oracle that always chooses the best performing baseline. These experiments show that even in an online setting, where the agent can leverage reward information and a Markov ground state when building its abstract state representation, explicitly encouraging Markov abstractions improves learning performance over state-of-the-art image-based RL.

# 8   Conclusion

We have developed a principled approach to learning abstract state representations that provably results in Markov abstract states, and which does not require estimating transition dynamics nor ground-state prediction. We defined what it means for a state abstraction to be Markov while ensuring that the abstract MDP accurately reflects the dynamics of the ground MDP, and introduced sufficient conditions for achieving such an abstraction. We adapted these conditions into a practical training objective that combines inverse model estimation and temporal contrastive learning. Our approach learns abstract state representations, with and without reward, that capture the structure of the underlying domain and substantially improve learning performance over existing approaches.

---

[8]We ran another experiment with no data augmentation, using a different state-of-the-art continuous control algorithm, RBF-DQN (Asadi et al., 2021), and found similar results there as well (see Appendix K for details).

## Acknowledgments and Disclosure of Funding

Thank you to Ben Abbatematteo, David Abel, Barrett Ames, Séb Arnold, Kavosh Asadi, Akhil Bagaria, Jake Beck, Jules Becker, Alexander Ivanov, Steve James, Michael Littman, Sam Lobel, and our other colleagues at Brown for thoughtful advice and countless helpful discussions, as well as Amy Zhang for generously sharing baseline learning curve data, and the anonymous ICML'21 and NeurIPS'21 reviewers for their time, consideration, and valuable feedback that substantially improved the paper. This research was supported by the ONR under the PERISCOPE MURI Contract N00014-17-1-2699, by the DARPA Lifelong Learning Machines program under grant FA8750-18-2-0117, and by NSF grants 1955361, 1717569, and CAREER award 1844960.

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
