# Appendix

# A Broader Impact Statement

Our approach enables agents to automatically construct useful, Markov state representations for reinforcement learning from rich observations. Since most reinforcement learning algorithms implicitly assume Markov abstract state representations, and since agents may struggle to learn when that assumption is violated, this work has the potential to benefit a large number of algorithms.

Our training objective is designed for neural networks, which are not guaranteed to converge to a global optimum when trained with stochastic gradient descent. Typically, such training objectives will only approximately satisfy the theoretical conditions they encode. However, this is not a drawback of our method—it applies to any representation learning technique that uses neural networks. Moreover, as neural network optimization techniques improve, our method will converge to a Markov abstraction, whereas other approaches may not. In the meantime, systems in safety-critical domains should ensure that they can cope with non-Markov abstractions without undergoing catastrophic failures.

We have shown experimentally that our method is effective in a variety of domains; however, other problem domains may require additional hyperparameter tuning, which can be expensive. Nevertheless, one benefit of our method is that Markov abstractions can be learned offline, without access to reward information. This means our algorithm could be used, in advance, to learn an abstraction for some problem domain, and then subsequent tasks in that environment (perhaps with different reward functions) could avoid the problem of perceptual abstraction as well as any associated training costs.

# B Glossary of Symbols

Here we provide a glossary of the most commonly-used symbols appearing in the rest of the paper.

| Symbol | Description |
|---|---|
| $M = (X, A, R, T, \gamma)$ | Ground MDP |
| $X$ | Observed (ground) state space |
| $A$ | Action space |
| $R(x', a, x)$ | Reward function |
| $T(x'\|a, x)$ | Transition model |
| $\gamma$ | Discount factor |
| $R^{(k)}(x', a, x, \{\cdots\}_{i=1}^{k})$ | Reward function conditioned on $k$ steps of additional history |
| $T^{(k)}(x'\|a, x, \{\cdots\}_{i=1}^{k})$ | Transition model conditioned on $k$ steps of additional history |
| $S$ | Unobserved (latent) state space, assumed for Block MDPs |
| $\sigma : S \to \Pr(X)$ | Sensor function for producing observed states |
| $\sigma^{-1} : X \to S$ | Hypothetical "inverse-sensor" function, assumed for Block MDPs |
| $\phi : X \to Z$ | Abstraction function |
| $w(x)$ | Fixed ground-state weighting function for constructing an abstract MDP (Li et al., 2006) |
| $B_{\phi,t}^{\pi}(x\|z_t)$ | Belief distribution for assigning policy- and time-dependent weights to ground states |
| $B_{\phi,t}^{\pi(k)}(x\|z_t, \{a_{t-i}, z_{t-i}\}_{i=1}^{k})$ | Belief distribution conditioned on $k$ steps of history |
| $M_\phi = (Z, A, R_{\phi,t}^{\pi}, T_{\phi,t}^{\pi}, \gamma)$ | Abstract decision process (possibly non-Markov) |
| $Z$ | Abstract state space |
| $R_{\phi,t}^{\pi}(z', a, z)$ | Abstract reward function |
| $T_{\phi,t}^{\pi}(z'\|a, z)$ | Abstract transition model |
| $R_{\phi,t}^{\pi(k)}(z', a, z, \{\cdots\}_{i=1}^{k})$ | Abstract reward function conditioned on $k$ steps of additional history |
| $T_{\phi,t}^{\pi(k)}(z'\|a, z, \{\cdots\}_{i=1}^{k})$ | Abstract transition model conditioned on $k$ steps of additional history |
| $\pi(a\|x)$ | A policy |
| $\pi^*$ | An optimal policy |
| $\Pi_C$ | An arbitrary policy class |
| $\Pi_\phi$ | The class of abstract policies induced by abstraction $\phi$ |
| $V^{\pi} : X \to \mathbb{R}$ | The value function induced by $\pi$ |
| $P_t^{\pi}(x)$ | Ground-state visitation distribution |
| $P_t^{\pi}(x'\|x)$ | Expected next-state dynamics model |
| $I_t^{\pi}(a\|x', x)$ | Inverse dynamics model |
| $P_{\phi,t}^{\pi}(z)$ | Abstract-state visitation distribution |
| $P_{\phi,t}^{\pi}(z'\|z)$ | Abstract expected next-state dynamics model |
| $I_{\phi,t}^{\pi}(a\|z', z)$ | Abstract inverse dynamics model |
| $P_t^{\pi}(x, a\|x')$ | Backwards dynamics model, used for kinematic inseparability |

Table 1: Glossary of symbols

# C   General-Policy Definitions

## C.1   General-Policy Definitions

The definitions of $T_{\phi,t}^{\pi}$ and $R_{\phi,t}^{\pi}$ in Section 4 only apply when the policy $\pi$ is a member of $\Pi_{\phi}$. Here we derive more general definitions that are valid for arbitrary policies, not just those in $\Pi_{\phi}$. For the special case where $\pi \in \Pi_{\phi}$, these definitions are equivalent to equations (4) and (5).

**Abstract transition probabilities.**

$$
\begin{aligned}
&\Pr(z'|a, z) \\
&= \sum_{x' \in X} \Pr(z'|x', a, z) \Pr(x'|a, z) \\
&= \sum_{x' \in X : \phi(x') = z'} \Pr(x'|a, z) \\
&= \sum_{x' \in X : \phi(x') = z'} \sum_{x \in X} \Pr(x'|a, x, z) \Pr(x|a, z) \\
&= \sum_{x' \in X : \phi(x') = z'} \sum_{x \in X} \Pr(x'|a, x, z) \frac{\Pr(a|x, z) \Pr(x|z)}{\sum_{\tilde{x} \in X} \Pr(a|\tilde{x}, z) \Pr(\tilde{x}_t|z_t)} \\
T_{\phi,t}^{\pi}(z'|a, z) := &\sum_{x' \in X : \phi(x') = z'} \sum_{x \in X : \phi(x) = z} T(x'|a, x) \frac{\pi_t(a|x) B_{\phi,t}^{\pi}(x|z)}{\sum_{\tilde{x} \in X} \pi_t(a|\tilde{x}) B_{\phi,t}^{\pi}(\tilde{x}|z)}
\end{aligned}
$$

**Abstract rewards.**

$$
\begin{aligned}
&\sum_{r \in R} r \Pr(r|z', a, z) \\
&= \sum_{x' \in X} \sum_{x \in X} \sum_{r \in R} r \Pr(r, x', x|z', a, z) \\
&= \sum_{x' \in X} \sum_{x \in X} \sum_{r \in R} r \Pr(r|x', z', a, x, z) \Pr(x', x|z', a, z) \\
&= \sum_{x' \in X} \sum_{x \in X} R(x', a, x) \frac{\Pr(z'|x', a, x, z) \Pr(x', x|a, z)}{\Pr(z'|a, z)} \\
&= \sum_{x' \in X : \phi(x') = z'} \sum_{x \in X} R(x', a, x) \frac{\Pr(x'|a, x, z) \Pr(x|a, z)}{\Pr(z'|a, z)} \\
&= \sum_{x' \in X : \phi(x') = z'} \sum_{x \in X} R(x', a, x) \frac{\Pr(x'|a, x)}{\Pr(z'|a, z)} \frac{\Pr(a|x) \Pr(x|z)}{\sum_{\tilde{x} \in X} \Pr(a|\tilde{x}) \Pr(\tilde{x}|z)} \\
R_{\phi,t}^{\pi}(z', a, z) := &\sum_{x' \in X : \phi(x') = z'} \sum_{x \in X : \phi(x) = z} R(x', a, x) \frac{T(x'|a, x) \pi_t(a|x) B_{\phi,t}^{\pi}(x|z)}{T_{\phi,t}^{\pi}(z'|a, z) \sum_{\tilde{x} \in X} \pi_t(a|\tilde{x}) B_{\phi,t}^{\pi}(\tilde{x}|z)}
\end{aligned}
$$

# D    Proofs

Here we provide proofs of Theorem 1 and its corollary, which state that the Inverse Model and Density Ratio conditions are sufficient for $\phi$ and $M_\phi$ to be Markov. Then, to complement the counterexample from Section 4.2, we also present and prove a second theorem which states that, given the Inverse Model condition, the Density Ratio condition is in fact *necessary* for a Markov abstraction.

## D.1    Main Theorem

The proof of Theorem 1 makes use of two lemmas: Lemma D.1, that equal $k$-step and $(k-1)$-step belief distributions imply equal $k$-step and $(k-1)$-step transition models and reward functions, and Lemma D.2, that equal $k$-step and $(k-1)$-step belief distributions imply equal $(k+1)$-step and $k$-step belief distributions. Since the lemmas apply for any arbitrary policy, we use the general-policy definitions from Appendix C.

**Lemma D.1.** *Given an MDP $M$, abstraction $\phi$, policy $\pi$, initial state distribution $P_0$, and any $k \geq 1$, if $B_{\phi,t}^{\pi(k)}(x_t|z_t, \{a_{t-i}, z_{t-i}\}_{i=1}^k) = B_{\phi,t}^{\pi(k-1)}(x_t|z_t, \{a_{t-i}, z_{t-i}\}_{i=1}^{k-1})$, then for all $a_t \in A$ and $z_{t+1} \in Z$:*

$$T_{\phi,t}^{\pi(k)}(z_{t+1}|\{a_{t-i}, z_{t-i}\}_{i=0}^k) = T_{\phi,t}^{\pi(k-1)}(z_{t+1}|\{a_{t-i}, z_{t-i}\}_{i=0}^{k-1})$$

$$\cap \quad R_{\phi,t}^{\pi(k)}(z_{t+1}, \{a_{t-i}, z_{t-i}\}_{i=0}^k) = R_{\phi,t}^{\pi(k-1)}(z_{t+1}, \{a_{t-i}, z_{t-i}\}_{i=0}^{k-1}).$$

In the proof below, we start with $B_{\phi,t}^{\pi(k)} = B_{\phi,t}^{\pi(k-1)}$, and repeatedly multiply or divide both sides by the same quantity, or take the same summations of both sides, to obtain $T_{\phi,t}^{\pi(k)} = T_{\phi,t}^{\pi(k-1)}$, then apply the same process again, making use of the fact that $T_{\phi,t}^{\pi(k)} = T_{\phi,t}^{\pi(k-1)}$, to obtain $R_{\phi,t}^{\pi(k)} = R_{\phi,t}^{\pi(k-1)}$.

**Proof:**

$$B_{\phi,t}^{\pi(k)}\left(x_t\big|z_t, \{a_{t-i}, z_{t-i}\}_{i=1}^k\right) = B_{\phi,t}^{\pi(k-1)}\left(x_t\big|z_t, \{a_{t-i}, z_{t-i}\}_{i=1}^{k-1}\right).$$

Let $a_t \in A$ be any action.

$$\Rightarrow \quad \pi_t(a_t|x_t)B_{\phi,t}^{\pi(k)}\left(x_t\big|z_t, \{a_{t-i}, z_{t-i}\}_{i=1}^k\right) = \pi_t(a_t|x_t)B_{\phi,t}^{\pi(k-1)}\left(x_t\big|z_t, \{a_{t-i}, z_{t-i}\}_{i=1}^{k-1}\right)$$

$$\Rightarrow \quad \frac{\pi_t(a_t|x_t)B_{\phi,t}^{\pi(k)}\left(x_t\big|z_t, \{a_{t-i}, z_{t-i}\}_{i=1}^k\right)}{\sum_{\tilde{x}_t \in X} \pi_t(a_t|\tilde{x}_t)B_{\phi,t}^{\pi(k)}\left(\tilde{x}_t\big|z_t, \{a_{t-i}, z_{t-i}\}_{i=1}^k\right)} = \frac{\pi_t(a_t|x_t)B_{\phi,t}^{\pi(k-1)}\left(x_t\big|z_t, \{a_{t-i}, z_{t-i}\}_{i=1}^{k-1}\right)}{\sum_{\tilde{x}_t \in X} \pi_t(a_t|\tilde{x}_t)B_{\phi,t}^{\pi(k-1)}\left(\tilde{x}_t\big|z_t, \{a_{t-i}, z_{t-i}\}_{i=1}^{k-1}\right)}. \tag{6}$$

Let

$$C_{\phi,t}^{\pi(k)} := \frac{\pi_t(a_t|x_t)B_{\phi,t}^{\pi(k)}\left(x_t|z_t, \{a_{t-i}, z_{t-i}\}_{i=1}^k\right)}{\sum_{\tilde{x}_t \in X} \pi_t(a_t|\tilde{x}_t)B_{\phi,t}^{\pi(k)}\left(\tilde{x}_t|z_t, \{a_{t-i}, z_{t-i}\}_{i=1}^k\right)} \tag{7}$$

Combining (6) and (7), we obtain:

$$C_{\phi,t}^{\pi(k)} = C_{\phi,t}^{\pi(k-1)} \tag{8}$$

$$\Rightarrow \quad \sum_{x_t \in z_t} \sum_{x_{t+1} \in z_{t+1}} T\left(x_{t+1} \mid a_t, x_t\right) C_{\phi,t}^{\pi(k)} = \sum_{x_t \in z_t} \sum_{x_{t+1} \in z_{t+1}} T\left(x_{t+1} \mid a_t, x_t\right) C_{\phi,t}^{\pi(k-1)}$$

$$\Leftrightarrow \quad T_{\phi,t}^{\pi(k)}\left(z_{t+1} \mid \{a_{t-i}, z_{t-i}\}_{i=0}^k\right) = T_{\phi,t}^{\pi(k-1)}\left(z_{t+1} \mid \{a_{t-i}, z_{t-i}\}_{i=0}^{k-1}\right). \tag{9}$$

Additionally, we can combine (8) and (9) and apply the same approach for rewards:

$$C_{\phi,t}^{\pi(k)} = C_{\phi,t}^{\pi(k-1)}$$

$$\Rightarrow \quad \frac{C_{\phi,t}^{\pi(k)}}{T_{\phi,t}^{\pi(k)}(z_{t+1}|\{a_{t-i}, z_{t-i}\}_{i=0}^k)} = \frac{C_{\phi,t}^{\pi(k-1)}}{T_{\phi,t}^{\pi(k-1)}(z_{t+1}|\{a_{t-i}, z_{t-i}\}_{i=0}^{k-1})}$$

$$\Rightarrow \quad \frac{T(x_{t+1}|a_t, z_t)C_{\phi,t}^{\pi(k)}}{T_{\phi,t}^{\pi(k)}(z_{t+1}|\{a_{t-i}, z_{t-i}\}_{i=0}^k)} = \frac{T(x_{t+1}|a_t, z_t)C_{\phi,t}^{\pi(k-1)}}{T_{\phi,t}^{\pi(k-1)}(z_{t+1}|\{a_{t-i}, z_{t-i}\}_{i=0}^{k-1})}$$

$$\Rightarrow \quad R_{\phi,t}^{\pi(k)}\left(z_{t+1}, \{a_{t-i}, z_{t-i}\}_{i=0}^k\right) = R_{\phi,t}^{\pi(k-1)}\left(z_{t+1}, \{a_{t-i}, z_{t-i}\}_{i=0}^{k-1}\right). \tag{10}$$

$$\square$$

**Lemma D.2.** *Given an MDP $M$, abstraction $\phi$, policy $\pi$, and initial state distribution $P_0$, if for all $t \geq k$, $z_t \in Z$, and $x_t \in X$ such that $\phi(x_t) = z_t$, it holds that $B_{\phi,t}^{\pi(k)}(x_t|z_t, \{a_{t-i}, z_{t-i}\}_{i=1}^{k}) = B_{\phi,t}^{\pi(k-1)}(x_t|z_t, \{a_{t-i}, z_{t-i}\}_{i=1}^{k-1})$, then for all $z_{t+1} \in Z$, $x_{t+1} \in X$ : $\phi(x_{t+1}) = z_{t+1}$,*

$$B_{\phi,t}^{\pi(k+1)}\left(x_{t+1}|z_{t+1}, \{a_{t-i}, z_{t-i}\}_{i=0}^{k}\right) = B_{\phi,t}^{\pi(k)}\left(x_{t+1}|z_{t+1}, \{a_{t-i}, z_{t-i}\}_{i=0}^{k-1}\right).$$

To prove this lemma, we invoke Lemma D.1 to obtain $T_{\phi,t}^{\pi(k)} = T_{\phi,t}^{\pi(k-1)}$, and then follow the same approach as before, performing operations to both sides until we achieve the desired result.

**Proof:**

Let $T_{\phi,t}^{\pi(k)}$ be defined via (4) and (3). Applying Lemma D.1 to the premise gives:

$$T_{\phi,t}^{\pi(k)}(z_{t+1}|\{a_{t-i}, z_{t-i}\}_{i=0}^{k}) = T_{\phi,t}^{\pi(k-1)}(z_{t+1}|\{a_{t-i}, z_{t-i}\}_{i=0}^{k-1}).$$

Returning to the premise, we have:

$$B_{\phi,t}^{\pi(k)}\left(x_t|z_t, \{a_{t-i}, z_{t-i}\}_{i=1}^{k}\right) = B_{\phi,t}^{\pi(k-1)}\left(x_t|z_t, \{a_{t-i}, z_{t-i}\}_{i=1}^{k-1}\right)$$

$$\Rightarrow \quad \frac{B_{\phi,t}^{\pi(k)}\left(x_t|z_t, \{a_{t-i}, z_{t-i}\}_{i=1}^{k}\right)}{T_{\phi,t}^{\pi(k)}\left(z_{t+1}|\{a_{t-i}, z_{t-i}\}_{i=0}^{k}\right)} = \frac{B_{\phi,t}^{\pi(k-1)}\left(x_t|z_t, \{a_{t-i}, z_{t-i}\}_{i=1}^{k-1}\right)}{T_{\phi,t}^{\pi(k-1)}\left(z_{t+1}|\{a_{t-i}, z_{t-i}\}_{i=0}^{k-1}\right)}$$

$$\Rightarrow \sum_{\substack{x_t \in X: \\ \phi(x_t) = z_t}} \frac{T(x_{t+1}|a_t, x_t) B_{\phi,t}^{\pi(k)}\left(x_t|z_t, \{a_{t-i}, z_{t-i}\}_{i=1}^{k}\right)}{T_{\phi,t}^{\pi(k)}\left(z_{t+1}|\{a_{t-i}, z_{t-i}\}_{i=0}^{k}\right)} = \sum_{\substack{x_t \in X: \\ \phi(x_t) = z_t}} \frac{T(x_{t+1}|a_t, x_t) B_{\phi,t}^{\pi(k-1)}\left(x_t|z_t, \{a_{t-i}, z_{t-i}\}_{i=1}^{k-1}\right)}{T_{\phi,t}^{\pi(k-1)}\left(z_{t+1}|\{a_{t-i}, z_{t-i}\}_{i=0}^{k-1}\right)}$$

$$\Rightarrow \quad B_{\phi,t}^{\pi(k+1)}\left(x_{t+1}|z_{t+1}, \{a_{t-i}, z_{t-i}\}_{i=0}^{k}\right) = B_{\phi,t}^{\pi(k)}\left(x_{t+1}|z_{t+1}, \{a_{t-i}, z_{t-i}\}_{i=0}^{k-1}\right).$$

$\square$

We now summarize the proof of the main theorem. We begin by showing that the belief distributions $B_{\phi,t}^{\pi(k)}$ and $B_{\phi,t}^{\pi(k-1)}$ must be equal for $k = 1$, and use Lemma D.1 to prove the base case of the theorem. Then we use Lemma D.2 to prove that the theorem holds in general.

**Proof of Theorem 1:**

**Base case.** For each $\pi \in \Pi_\phi$, let $B_{\phi,t}^{\pi}$ be defined via (2). Then, starting from the Density Ratio condition, for any $z_{t+1}, z_t \in Z$, and $x_{t+1} \in X$ such that $\phi(x_{t+1}) = z_{t+1}$, and any action $a_{t-1} \in A$:

$$\frac{P_{\phi,t}^{\pi}(z_t|z_{t-1})}{P_{\phi,t}^{\pi}(z_t)} = \frac{P_t^{\pi}(x_t|z_{t-1})}{P_t^{\pi}(x_t)}$$

$$\Rightarrow \quad \frac{P_{\phi,t}^{\pi}(z_t|z_{t-1})}{P_{\phi,t}^{\pi}(z_t)} = \sum_{x_{t-1} \in X} \frac{P_t^{\pi}(x_t|x_{t-1})}{P_t^{\pi}(x_t)} B_{\phi,t}^{\pi}(x_{t-1}|z_{t-1})$$

$$\Rightarrow \quad \frac{P_t^{\pi}(x_t)}{P_{\phi,t}^{\pi}(z_t)} = \sum_{x_{t-1} \in X} \frac{P_t^{\pi}(x_t|x_{t-1}) B_{\phi,t}^{\pi}(x_{t-1}|z_{t-1})}{P_{\phi,t}^{\pi}(z_t|z_{t-1})} \cdot \frac{I_t^{\pi}(a_{t-1}|x_t, x_{t-1}) \pi_{\phi,t}(a_{t-1}|z_{t-1})}{I_{\phi,t}^{\pi}(a_{t-1}|z_t, z_{t-1}) \pi_t(a_{t-1}|x_{t-1})}$$

$$\Rightarrow \quad \frac{\mathbb{1}[\phi(x_t) = z_t] \, P_t^{\pi}(x_t)}{\sum_{\tilde{x}_t \in X} P_t^{\pi}(\tilde{x}_t)} = \mathbb{1}[\phi(x_t) = z_t] \sum_{x_{t-1} \in X} \frac{T(x_t|a_{t-1}, x_{t-1}) B_{\phi,t}^{\pi}(x_{t-1}|z_{t-1})}{T_{\phi,t}^{\pi}(z_t|a_{t-1}, z_{t-1})}$$

$$\Rightarrow \quad B_{\phi,t}^{\pi}(x_t|z_t) = \frac{\mathbb{1}[\phi(x_t) = z_t] \sum_{x_{t-1} \in X} T_\phi(x_t|a_{t-1}, x_{t-1}) B_{\phi,t}^{\pi}(x_{t-1}|z_{t-1})}{\sum_{\tilde{x}_t \in z_t} \sum_{\tilde{x}_{t-1} \in z_{t-1}} T(\tilde{x}_t|a_{t-1}, \tilde{x}_{t-1}) B_{\phi,t}^{\pi}(\tilde{x}_{t-1}|z_{t-1})}$$

$$\Rightarrow \quad B_{\phi,t}^{\pi(0)}(x_t|z_t) = B_{\phi,t}^{\pi(1)}(x_t|z_t, a_{t-1}, z_{t-1}) \tag{11}$$

Here (11) satisfies the conditions of Lemma D.1 (with $k = 1$), therefore, for all $a_t \in A$:

$$T_{\phi,t}^{\pi(0)}(z_{t+1}|a_t, z_t) = T_{\phi,t}^{\pi(1)}(z_{t+1}|a_t, z_t, a_{t-1}, z_{t-1})$$

$$\text{and} \quad R_{\phi,t}^{\pi(0)}(z_{t+1}, a_t, z_t) = R_{\phi,t}^{\pi(1)}(z_{t+1}, a_t, z_t, a_{t-1}, z_{t-1})$$

This proves the theorem for $k = 1$.

**Induction on $k$.** Note that (11) also allows us to apply Lemma D.2. Therefore, by induction on $k$:

$$B_{\phi,t}^{\pi(k+1)}\left(x_{t+1}\big|z_{t+1}, \{a_{t-i}, z_{t-i}\}_{i=0}^{k}\right) = B_{\phi,t}^{\pi}(x_{t+1}|z_{t+1}) \quad \forall\, k \geq 1 \tag{12}$$

$\square$

When (12) holds, we informally say that the belief distribution is Markov.

**Proof of Corollary 1.1:** Follows directly from Definition 2 and Lemma D.1 via induction on $k$. $\square$

This first corollary says that a Markov abstraction implies a Markov abstract state representation. The next one says that, if a belief distribution is non-Markov over some horizon $n$, it must also be non-Markov when conditioning on a single additional timestep.

**Corollary 1.2.** *If there exists some $n \geq 1$ such that $B_{\phi,t}^{\pi(n)} \neq B_{\phi,t}^{\pi}$, then $B_{\phi,t}^{\pi(1)} \neq B_{\phi,t}^{\pi}$.*

**Proof:** Suppose such an $n$ exists, and assume for the sake of contradiction that $B_{\phi,t}^{\pi(1)} = B_{\phi,t}^{\pi}$. Then by Lemma D.2, $B_{\phi,t}^{\pi(k)} = B_{\phi,t}^{\pi}$ for all $k \geq 1$. However this is impossible, since we know there exists some $n \geq 1$ such that $B_{\phi,t}^{\pi(n)} \neq B_{\phi,t}^{\pi}$. Therefore $B_{\phi,t}^{\pi(1)} \neq B_{\phi,t}^{\pi}$. $\square$

### D.2 Inverse Model Implies Density Ratio

As discussed in Section 4.2, the Inverse Model condition is not sufficient to ensure a Markov abstraction. In fact, what is missing is precisely the Density Ratio condition. Theorem 1 already states that, given the Inverse Model condition, the Density Ratio condition is sufficient for an abstraction to be Markov over its policy class; the following theorem states that it is also necessary.

**Theorem 2.** *If $\phi : X \to Z$ is a Markov abstraction of MDP $M = (X, A, R, T, \gamma)$ for any policy in the policy class $\Pi_\phi$, and the Inverse Model condition of Theorem 1 holds for every timestep $t$, then the Density Ratio condition also holds for every timestep $t$.*

**Proof:**

Since $\phi$ is a Markov abstraction, equation (12) holds for any $k \geq 1$. Fixing $k = 1$, we obtain:

$$B_{\phi,t}^{\pi(1)}(x'|z', a, z) = B_{\phi,t}^{\pi(0)}(x'|z')$$

$$\frac{\mathbb{1}[\phi(x') = z']\sum_{\tilde{x}\in X} T(x'|a, \tilde{x})B_{\phi,t}^{\pi}(\tilde{x}|z)}{\sum_{\tilde{x}'\in X:\phi(x')=z'}\sum_{\tilde{x}\in X:\phi(x)=z} T(\tilde{x}'|a, \tilde{x})B_{\phi,t}^{\pi}(\tilde{x}|z)} = \frac{\mathbb{1}[\phi(x') = z']P_t^{\pi}(x')}{\sum_{\tilde{x}'\in X:\phi(\tilde{x}')=z'} P_t^{\pi}(\tilde{x}')}$$

$$\frac{\sum_{\tilde{x}\in X} T(x'|a, \tilde{x})B_{\phi,t}^{\pi}(\tilde{x}|z)}{T_{\phi,t}^{\pi}(z'|a, z)} = \frac{P_t^{\pi}(x')}{P_{\phi,t}^{\pi}(z')}$$

$$\frac{\sum_{\tilde{x}\in X} T(x'|a, \tilde{x})B_{\phi,t}^{\pi}(\tilde{x}|z)}{P_t^{\pi}(x')} = \frac{T_{\phi,t}^{\pi}(z'|a, z)}{P_{\phi,t}^{\pi}(z')}$$

$$\sum_{\tilde{x}\in X} \frac{I_t^{\pi}(a|x', \tilde{x})P_t^{\pi}(x'|\tilde{x})B_{\phi,t}^{\pi}(\tilde{x}|z)}{P_t^{\pi}(x')\pi(a|\tilde{x})} = \frac{I_{\phi,t}^{\pi}(a|z', z)P_{\phi,t}^{\pi}(z'|z)}{P_{\phi,t}^{\pi}(z')\pi(a|z)}$$

$$\frac{\sum_{\tilde{x}\in X} I_t^{\pi}(a|x', \tilde{x})P_t^{\pi}(x'|\tilde{x})B_{\phi,t}^{\pi}(\tilde{x}|z)}{P_t^{\pi}(x')} = \frac{I_{\phi,t}^{\pi}(a|z', z)P_{\phi,t}^{\pi}(z'|z)}{P_{\phi,t}^{\pi}(z')} \tag{13}$$

Here we apply the Inverse Model condition, namely that $I_t^{\pi}(a|x', x) = I_{\phi,t}^{\pi}(a|z', z)$ for all $z, z' \in Z$; $x, x' \in X$, such that $\phi(x') = z'$ and $\phi(x) = z$.

$$\frac{\sum_{\tilde{x}\in X} P_t^{\pi}(x'|\tilde{x})B_{\phi,t}^{\pi}(\tilde{x}|z)}{P_t^{\pi}(x')} = \frac{P_{\phi,t}^{\pi}(z'|z)}{P_{\phi,t}^{\pi}(z')}$$

$$\frac{P_t^{\pi}(x'|z)}{P_t^{\pi}(x')} = \frac{P_{\phi,t}^{\pi}(z'|z)}{P_{\phi,t}^{\pi}(z')} \qquad \text{(Density Ratio)}$$

$\square$

It may appear that Theorems 1 and 2 together imply that the Inverse Model and Density Ratio conditions are necessary and sufficient for an abstraction to be Markov over its policy class; however, this is not quite true. Both conditions, taken together, are sufficient for an abstraction to be Markov, and, *given the Inverse Model condition*, the Density Ratio condition is necessary. Examining equation (13), we see that, had we instead assumed the Density Ratio condition for Theorem 2 (rather than the Inverse Model condition), we would not recover $I_t^\pi(a|x', x) = I_{\phi,t}^\pi(a|z', z)$, but rather $\sum_{\tilde{x} \in X} I_t^\pi(a|x', \tilde{x}) B_{\phi,t}^\pi(\tilde{x}|z) = I_{\phi,t}^\pi(a|z', z)$. That is, the Inverse Model condition would only be guaranteed to hold in expectation, but not for arbitrary $x \in X$.

# E Derivation of Density Ratio Objective

Our Density Ratio objective in Section 5 is based on the following derivation, adapted from Tiao (2017).

Suppose we have a dataset consisting of samples $X_c = \{x_c'^{(i)}\}_{i=1}^{n_c}$ drawn from conditional distribution $\Pr(x'|x)$, and samples $X_m = \{x_m'^{(j)}\}_{j=1}^{n_m}$ drawn from marginal distribution $\Pr(x')$. We assign label $y = 1$ to samples from $X_c$ and $y = 0$ to samples from $X_m$, and our goal is to predict the label associated with each sample. To construct an estimator, we rename the two distributions $p(x'|y = 1) := \Pr(x'|z)$ and $p(x'|y = 0) := \Pr(x')$ and rewrite the density ratio $\delta(x') := \frac{\Pr(x'|x)}{\Pr(x')}$ as follows:

$$\delta(x') = \frac{p(x'|y=1)}{p(x'|y=0)} = \frac{p(y=1|x')p(x')}{p(y=1)} \frac{p(y=0)}{p(y=0|x')p(x')} = \frac{n_m}{(n_m+n_c)} \frac{(n_m+n_c)}{n_c} \frac{p(y=1|x')}{p(y=0|x')} = \frac{n_m}{n_c} \frac{p(y=1|x')}{1-p(y=1|x')}. \tag{14}$$

When $n_c = n_m = N$, which is the case for our implementation, the leading fraction can be ignored. To estimate $\delta(x')$, we can simply train a classifier $g(x', x; \theta_g)$ to approximate $p(y = 1|x')$ and then substitute $g$ for $p(y = 1|x')$ in (14).

However, we need not estimate $\delta(x')$ to satisfy the Density Ratio condition; we need only ensure $\delta_\phi(z') = \mathbb{E}_{B_\phi}[\delta(x')]$. We therefore repeat the derivation for abstract states and obtain $\delta_\phi(z') := \frac{\Pr(z'|z)}{\Pr(z')} = \frac{n_m}{n_c} \frac{q(y=1|z')}{1-q(y=1|z')}$, where $q$ is our renamed distribution, and modify our classifier $g$ to accept abstract states instead of ground states. The labels are the same regardless of whether we use ground or abstract state pairs, so training will cause $g$ to approach $p$ and $q$ simultaneously, thus driving $\delta_\phi(z') \to \delta(x')$ and satisfying the Density Ratio condition.

## F Markov State Abstractions and Kinematic Inseparability

As discussed in Section 3, the notion of kinematic inseparability (Misra et al., 2020) is closely related to Markov abstraction. Recall that two states $x'_1$ and $x'_2$ are defined to be kinematically inseparable if $\Pr(x, a|x'_1) = \Pr(x, a|x'_2)$ and $T(x''|a, x'_1) = T(x''|a, x'_2)$ (which the authors call "backwards" and "forwards" KI, respectively). Misra et al. (2020) define kinematic inseparability abstractions over the set of all possible "roll-in" distributions $u(x, a)$ supported on $X \times A$, and technically, the backwards KI probabilities $\Pr(x, a|x')$ depend on $u$. However, to support choosing a policy class, we can just as easily define $u$ in terms of a policy: $u(x, a) := \pi(a|x)P_t^\pi(x)$. This formulation leads to:

$$P_t^\pi(x, a|x') := \frac{T(x'|a, x)\pi(a|x)P_t^\pi(x)}{\sum_{\tilde{x}\in X, \tilde{a}\in A} T(x'|\tilde{a}, \tilde{x})\pi(\tilde{a}|\tilde{x})P_t^\pi(\tilde{x})}.$$

**Definition 3.** *An abstraction $\phi : X \to Z$ is a* kinematic inseparability abstraction *of MDP $M = (X, A, R, T, \gamma)$ over policy class $\Pi_C$, if for all policies $\pi \in \Pi_C$, and all $a \in A; x, x'_1, x'_2, x'' \in X$ such that $\phi(x'_1) = \phi(x'_2)$; $P_t^\pi(x, a|x'_1) = P_t^\pi(x, a|x'_2)$ and $T(x''|a, x'_1) = T(x''|a, x'_2)$.*

Similarly, we can define forward—or backward—KI abstractions where only $T(x''|a, x'_1) = T(x''|a, x'_2)$—or respectively, $P_t^\pi(x, a|x'_1) = P_t^\pi(x, a|x'_2)$—is guaranteed to hold. A KI abstraction is one that is both forward KI and backward KI.

The KI conditions are slightly stronger conditions than those of Theorem 1, as the following example demonstrates.

### F.1 Example MDP

The figure below modifies the transition dynamics of the MDP in Section 2, such that the action $a_1$ has the same effect everywhere: to transition to either central state, $x_1$ or $x_2$, with equal probability.

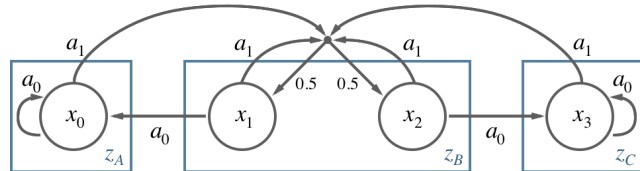

Figure 5: An MDP and a Markov abstraction that is not a KI abstraction.

By the same reasoning as in Section 4.2, the Inverse Model condition holds here, but now, due to the shared transition dynamics of action $a_1$, the Density Ratio condition holds as well, for any policy in $\Pi_\phi$. We can apply Theorem 1 to see that the abstraction is Markov, or we can simply observe that conditioning the belief distribution $B_{\phi,t}^\pi(x|z_B)$ on additional history has no effect, since any possible trajectory ending in $z_B$ leads to the same 50–50 distribution over ground states $x_1$ and $x_2$. Either way, $\phi$ is Markov by Definition 2.

This abstraction also happens to satisfy the backwards KI condition, since $P_t^\pi(x, a|x'_1) = P_t^\pi(x, a|x'_2)$ for any $(x, a)$ pair and any policy. However, clearly $T(x'|a_0, x_1) \neq T(x'|a_0, x_2)$, and therefore the forwards KI condition does not hold and this is not a KI abstraction.

This example shows that the Markov conditions essentially take the stance that because $\pi$ is restricted to the policy class $\Pi_\phi$, knowing the difference between $x_1$ and $x_2$ doesn't help, because $\pi$ must have the same behavior for both ground states. By contrast, the KI conditions take the stance that because $x_1$ and $x_2$ have different dynamics, the agent may wish to change its behavior based on which state it sees, so it ought to choose an abstraction that does not limit decision making in that respect.

### F.2 "Strongly Markov" Implies KI

In Section 5, we mentioned that the Density Ratio training objective was stronger than necessary to ensure the corresponding condition of Theorem 1. Instead of encoding the condition $\frac{\Pr(x'|z)}{\Pr(x')} = \frac{\Pr(z'|z)}{\Pr(z')}$, we discussed how the contrastive training procedure actually encodes the stronger condition $\frac{\Pr(x'|x)}{\Pr(x')} = \frac{\Pr(z'|z)}{\Pr(z')}$ that holds for each ground state $x$ individually, rather than just in expectation.

Let us call the latter condition the Strong Density Ratio condition, and call its combination with the Inverse Model condition the Strong Markov conditions.

Clearly the Strong Markov conditions imply the original Markov conditions, and, as the following theorem shows, they also imply the KI conditions.

**Theorem 3.** *If $\phi : X \to Z$ is an abstraction of MDP $M = (X, A, R, T, \gamma)$ such that for any policy $\pi$ in the policy class $\Pi_\phi$, both the Inverse Model condition of Theorem 1 and the Strong Density Ratio condition—i.e. $\frac{P_t^\pi(x'|x)}{P_t^\pi(x')} = \frac{P_{\phi,t}^\pi(z'|z)}{P_{\phi,t}^\pi(z')}$, for all $z, z' \in Z$; $x, x' \in X$ such that $\phi(x) = z$ and $\phi(x') = z'$—hold for every timestep $t$, then $\phi$ is a kinematic inseparability abstraction .*

**Proof:**

Starting from the Inverse Model condition, we have $I_t^\pi(a|x', x) = I_{\phi,t}^\pi(a|z', z)$ for all $z, z' \in Z$; $x, x' \in X$ such that $\phi(x) = z$ and $\phi(x') = z'$. Independently varying either $x' \in \phi^{-1}(z')$ or $x \in \phi^{-1}(z)$, we obtain the following:

$$[\text{Vary } x'] \to \quad I_t^\pi(a|x_1', x) = I_t^\pi(a|x_2', x) \tag{15}$$

$$[\text{Vary } x] \to \quad I_t^\pi(a|x', x_1) = I_t^\pi(a|x', x_2) \tag{16}$$

Similarly, if we start from the Strong Density Ratio condition, we obtain:

$$[\text{Vary } x'] \to \quad \frac{P_t^\pi(x_1'|x)}{P_t^\pi(x_1')} = \frac{P_t^\pi(x_2'|x)}{P_t^\pi(x_2')} \tag{17}$$

$$[\text{Vary } x] \to \quad \frac{P_t^\pi(x'|x_1)}{P_t^\pi(x')} = \frac{P_t^\pi(x'|x_2)}{P_t^\pi(x')} \tag{18}$$

If we apply Bayes' theorem to (17), we can cancel terms in the result, and also in (18), to obtain:

$$P_t^\pi(x|x_1') = P_t^\pi(x|x_2') \tag{19}$$

$$\text{and} \quad P_t^\pi(x'|x_1) = P_t^\pi(x'|x_2). \tag{20}$$

Combining (15) with (19), we obtain the backwards KI condition:

$$I_t^\pi(a|x_1', x)P_t^\pi(x|x_1') = I_t^\pi(a|x_2', x)P_t^\pi(x|x_2')$$

$$P_t^\pi(x, a|x_1') = P_t^\pi(x, a|x_2') \tag{Backwards KI}$$

Similarly, we can combine (16) with (20) to obtain the forwards KI condition:

$$I_t^\pi(a|x', x_1)P_t^\pi(x'|x_1) = I_t^\pi(a|x', x_2)P_t^\pi(x'|x_2)$$

$$T(x'|a, x_1)\pi(a|x_1) = T(x'|a, x_2)\pi(a|x_2)$$

$$T(x'|a, x_1) = T(x'|a, x_2) \tag{Forwards KI}$$

$$\square$$

Thus, we see that the training objectives in Section 5 encourage learning a kinematic inseparability abstraction in addition to a Markov abstraction. This helps avoid representation collapse by ensuring that we do not group together any states for which a meaningful kinematic distinction can be made.

# G Implementation Details for Visual Gridworld

The visual gridworld is a $6 \times 6$ grid with four discrete actions: up, down, left, and right. Observed states are generated by converting the agent's $(x, y)$ position to a one-hot image representation (see Figure 1, left). The image displays each position in the $6 \times 6$ grid as a 3px-by-3px patch, inside of which we light up one pixel (in the center) and then smooth it using a truncated Gaussian kernel. This results in an $18 \times 18$ image (where 3px-by-3px grid cells are equidistant), to which we then add per-pixel noise from another truncated Gaussian. During pretraining, there are no rewards or terminal states. During training, for each random seed, a single state is designated to be the goal state, and the agent receives $-1$ reward per timestep until it reaches the goal state, at which point a new episode begins, with the agent in a random non-goal location.

## G.1 Computing Resources

To build the figures in the paper, we pretrained 5 different abstractions, and trained 7 different agents, each with 300 seeds. Each 3000-step pretraining run takes about 1 minute, and each training run takes about 30 seconds, on a 2016 MacBook Pro 2GHz i5 with no GPU, for a total of about 42 compute hours. We ran these jobs on a computing cluster with comparable processors or better.

## G.2 Network Architectures

```
FeatureNet(
  (phi): Encoder(
    (0): Reshape(-1, 252)
    (1): Linear(in_features=252, out_features=32, bias=True)
    (2): Tanh()
    (3): Linear(in_features=32, out_features=2, bias=True)
    (4): Tanh()
  )
  (inv_model): InverseNet(
    (0): Linear(in_features=4, out_features=32, bias=True)
    (1): Tanh()
    (2): Linear(in_features=32, out_features=4, bias=True)
  )
  (contr_model): ContrastiveNet(
    (0): Linear(in_features=4, out_features=32, bias=True)
    (1): Tanh()
    (2): Linear(in_features=32, out_features=1, bias=True)
    (3): Sigmoid()
  )
)
QNet(
  (0): Linear(in_features=2, out_features=32, bias=True)
  (1): ReLU()
  (2): Linear(in_features=32, out_features=4, bias=True)
)
AutoEncoder / PixelPredictor(
  (phi): Encoder(
    (0): Reshape(-1, 252)
    (1): Linear(in_features=252, out_features=32, bias=True)
    (2): Tanh()
    (3): Linear(in_features=32, out_features=2, bias=True)
    (4): Tanh()
  )
  (phi_inverse): Decoder(
    (0): Linear(in_features=2, out_features=32, bias=True)
    (1): Tanh()
    (2): Linear(in_features=32, out_features=252, bias=True)
    (3): Tanh()
    (4): Reshape(-1, 21, 12)
  )
  MSELoss()
)
```

### G.3 Hyperparameters

We tuned the DQN hyperparameters until it learned effectively with expert features (i.e. ground-truth $(x, y)$ position), then we left the DQN hyperparameters fixed while tuning pretraining hyperparameters. For pretraining, we considered 3000 and 30,000 gradient updates (see Appendix J), and batch sizes within {512, 1024, 2048}. We found that the higher batch size was helpful for stabilizing the offline representations. We also did some informal experiments with latent dimensionality above 2, such as 3 or 10, which produced similar results: representations were still Markov, but harder to interpret. We use 2 dimensions in the paper for ease of visualization. We did not tune the loss coefficients, but we include ablations where either $\alpha$ or $\beta$ is set to zero.

| Hyperparameter | Value |
|---|---|
| Number of seeds | 300 |
| Optimizer | Adam |
| Learning rate | 0.003 |
| Batch size | 2048 |
| Gradient updates | 3000 |
| Latent dimensions | 2 |
| Number of conditional samples, $n_c$ | 1 |
| Number of marginal samples, $n_m$ | 1 |
| Loss coefficients | |
| $\mathcal{L}_{\text{Inverse}}$ $(\alpha)$ | 1.0 |
| $\mathcal{L}_{\text{Contrastive}}$ $(\beta)$ | 1.0 |
| $\mathcal{L}_{\text{Smoothness}}$ $(\eta)$ | 0.0 |

Table 2: Pretraining hyperparameters

| Hyperparameter | Value |
|---|---|
| Number of seeds | 300 |
| Number of episodes | 100 |
| Maximum steps per episode | 1000 |
| Optimizer | Adam |
| Learning rate | 0.003 |
| Batch size | 16 |
| Discount factor, $\gamma$ | 0.9 |
| Starting exploration probability, $\epsilon_0$ | 1.0 |
| Final exploration probability, $\epsilon$ | 0.05 |
| Epsilon decay period | 2500 |
| Replay buffer size | 10000 |
| Initialization steps | 500 |
| Target network copy period | 50 |

Table 3: DQN hyperparameters

# H   Implementation Details for DeepMind Control

We use the same RAD network architecture and code implementation as Laskin et al. (2020a), which we customized to add our Markov objective. We note that there was a discrepancy between the batch size in their code implementation (128) and what was reported in the original paper (512); we chose the former for our experiments.

The SAC (expert) results used the code implementation from Yarats & Kostrikov (2020).

The DBC, DeepMDP, CPC, and SAC-AE results are from Zhang et al. (2021), except for Ball_in_Cup, which they did not include in their experimental evaluation. We ran their DBC code independently (with the same settings they used) to produce our Ball_in_Cup results.

## H.1   Computing Resources

To build the graph in Section 7, we trained 4 agents on 6 domains, with 10 seeds each. Each training run (to 500,000 steps) takes between 24 and 36 hours (depending on the action repeat for that domain), on a machine with two Intel Xeon Gold 5122 vCPUs and shared access to one Nvidia 1080Ti GPU, for a total of approximately 7200 compute hours. We ran these jobs on a computing cluster with comparable hardware or better.

## H.2   Markov Network Architecture

```
MarkovHead(
  InverseModel(
    (body): Sequential(
      (0): Linear(in_features=100, out_features=1024, bias=True)
      (1): ReLU()
      (2): Linear(in_features=1024, out_features=1024, bias=True)
      (3): ReLU()
    )
    (mean_linear): Linear(in_features=1024, out_features=action_dim, bias
        =True)
    (log_std_linear): Linear(in_features=1024, out_features=action_dim,
        bias=True)
  )
  ContrastiveModel(
    (model): Sequential(
      (0): Linear(in_features=100, out_features=1024, bias=True)
      (1): ReLU()
      (2): Linear(in_features=1024, out_features=1, bias=True)
    )
  )
  BCEWithLogitsLoss()
)
```

## H.3   Hyperparameters

When tuning our algorithm, we left all RAD hyperparameters fixed except init_steps, which we increased to equal 10 episodes across all domains to provide adequate coverage for pretraining. We compensated for this change by adding (init_steps - 1K) catchup learning steps to ensure both methods had the same number of RL updates. This means our method is at a slight disadvantage, since RAD can begin learning from reward information after just 1K steps, but our method must wait until after the first 10 episodes of uniform random exploration. Otherwise, we only considered changes to the Markov hyperparameters (see Table 5). We set the Markov learning rate equal to the RAD learning rate for each domain (and additionally considered 5e-5 for cheetah only). We tuned the $\mathcal{L}_{Inv}$ loss coefficient within {0.1, 1.0, 10.0, 30.0}, and the $\mathcal{L}_{Smooth}$ loss coefficient within {0, 10.0, 30.0}. The other hyperparameters, including the network architecture, we did not change from their initial values.

| Hyperparameter | Value |
|---|---|
| Augmentation | |
|    Walker | Crop |
|    Others | Translate |
| Observation rendering | (100, 100) |
| Crop size | (84, 84) |
| Translate size | (108, 108) |
| Replay buffer size | 100000 |
| Initial steps | 1000 |
| Stacked frames | 3 |
| Action repeat | 2; finger, |
| |    walker |
| | 8; cartpole |
| | 4; others |
| Hidden units (MLP) | 1024 |
| Evaluation episodes | 10 |
| Optimizer | Adam |
|   $(\beta_1, \beta_2) \rightarrow (\phi, \pi, Q)$ | (.9, .999) |
|   $(\beta_1, \beta_2) \rightarrow (\alpha)$ | (.5, .999) |
|   Learning rate $(\phi, \pi, Q)$ | 2e-4, cheetah |
| | 1e-3, others |
|   Learning rate $(\alpha)$ | 1e-4 |
| Batch size | 128 |
| Q function EMA $\tau$ | 0.01 |
| Critic target update freq | 2 |
| Convolutional layers | 4 |
| Number of filters | 32 |
| Non-linearity | ReLU |
| Encoder EMA $\tau$ | 0.05 |
| Latent dimension | 50 |
| Discount $\gamma$ | .99 |
| Initial temperature | 0.1 |

Table 4: RAD hyperparameters

| Hyperparameter | Value | |
|---|---|---|
| Pretraining steps | 100K | |
| Pretraining batch size | 512 | |
| RAD init steps | (20K / action_repeat) | |
| RAD catchup steps | (init_steps - 1K) | |
| Other RAD parameters | unchanged | |
| Loss coefficients | | |
|   $\mathcal{L}_{Inv}$ | 30.0 | ball, |
| | | reacher |
| | 1.0 | others |
|   $\mathcal{L}_{Ratio}$ | 1.0 | |
|   $\mathcal{L}_{Smooth}$ | 30.0 | ball, |
| | | reacher, |
| | | cheetah |
| | 10.0 | others |
| Smoothness $d_0$ | 0.01 | |
| Conditional samples, $n_c$ | 128 | |
| Marginal samples, $n_m$ | 128 | |
| Optimizer | Adam | |
|   $(\beta_1, \beta_2) \rightarrow$ (Markov) | (.9, .999) | |
|   Learning rate | 2e-4 | cheetah |
| | 1e-3 | others |

Table 5: Markov hyperparameters

# I  Additional Representation Visualizations

Here we visualize abstraction learning progress for the $6 \times 6$ visual gridworld domain for six random seeds. Each figure below displays selected frames (progressing from left to right) of a different abstraction learning method (top to bottom): $\mathcal{L}_{Markov}$; $\mathcal{L}_{Inv}$ only; $\mathcal{L}_{Ratio}$ only; autoencoder; pixel prediction. The networks are initialized identically for each random seed. Color denotes ground-truth $(x, y)$ position, which is not shown to the agent. These visualizations span 30,000 training steps (columns, left to right: after 1, 100, 200, 700, 3K, 10K, and 30K steps, respectively). In particular, the third column from the right shows the representations after 3000 steps, which we use for the results in the main text. We show additional learning curves for the final representations in Appendix J.

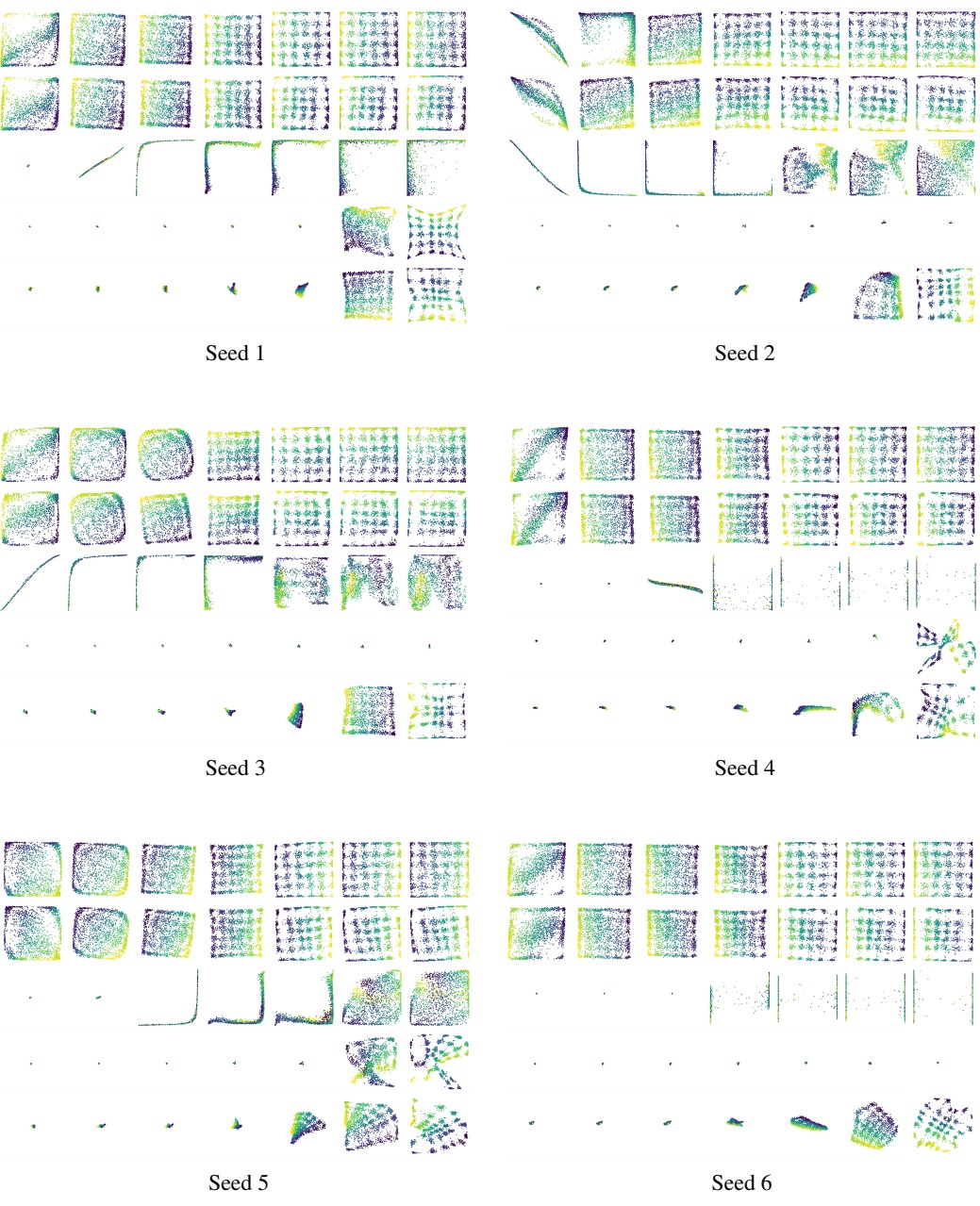

Seed 1    Seed 2

Seed 3    Seed 4

Seed 5    Seed 6

Figure 7

## J   Gridworld Results for Increased Pretraining Time

Since some of the representations in Appendix I appeared not to have converged after just 3000 training steps, we investigated whether the subsequent learning performance would improve with more pretraining. We found that increasing the number of pretraining steps from 3000 to 30,000 improves the learning performance of $\phi_{Ratio}$ and $\phi_{Autoenc}$ and $\phi_{PixelPred}$ (see Figure 8), with the latter representation now matching the performance of $\phi_{Markov}$.

It is perhaps unsurprising that the pixel prediction model eventually recovers the performance of the Markov abstraction, because the pixel prediction task is a valid way to ensure Markov abstract states. However, as we discuss in Sec. 3.2, the pixel prediction objective is misaligned with the basic goal of state abstraction, since it must effectively throw away no information. It is clear from Figures 7 and 8 that our method is able to reliably learn a Markov representation about ten times faster than pixel prediction, which reflects the fact that the latter is a fundamentally more challenging objective.

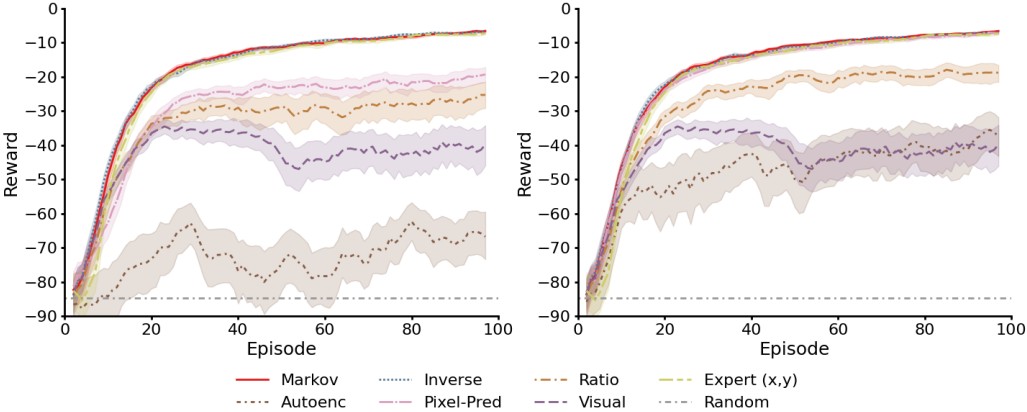

Figure 8: Mean episode reward for the visual gridworld navigation task, using representations that were pretrained for 3,000 steps (left) versus 30,000 steps (right). Increased pretraining time improves the performance of $\phi_{Ratio}$, $\phi_{Autoenc}$ and $\phi_{PixelPred}$. (300 seeds; 5-point moving average; shaded regions denote 95% confidence intervals.)

# K    DeepMind Control Experiment with RBF-DQN

Recently, Asadi et al. (2021) showed how to use radial basis functions for value-function based RL in problems with continuous action spaces. When trained with ground-truth state information, RBF-DQN achieved state-of-the-art performance on several continuous control tasks; however, to our knowledge, the algorithm has not yet been used for image-based domains.

We trained RBF-DQN from stacked image inputs on "Finger, Spin," one of the tasks from Section 7, customizing the authors' PyTorch implementation to add our Markov training objective. We do not use any data augmentation or smoothness loss, and we skip the pretraining phase entirely; we simply add the Markov objective as an auxiliary task during RL. Here we again observe that adding the Markov objective improves learning performance over the visual baseline and approaches the performance of using ground-truth state information (see Figure 9).

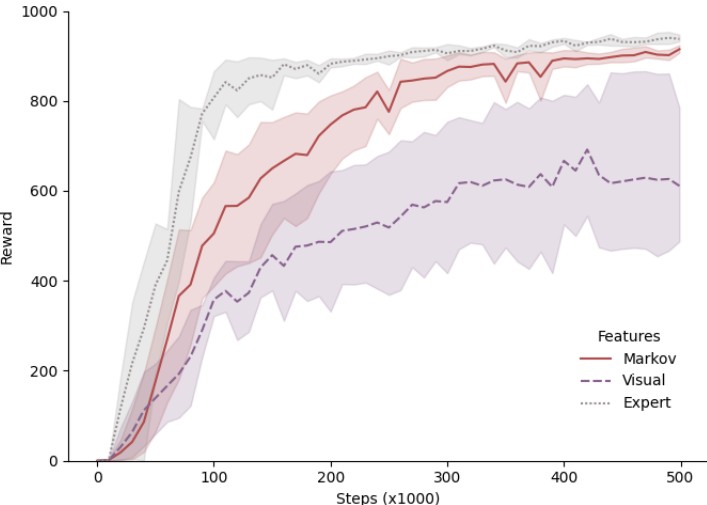

Figure 9: Mean episode reward for RBF-DQN on "Finger, Spin" with Markov, visual, and expert features. Adding the Markov objective dramatically improves performance over the visual baseline. (Markov – 5 seeds; Visual – 6 seeds; Expert – 3 seeds; shaded regions denote 95% confidence intervals).

## L   Investigating the Value of Smoothness in Markov Abstractions

Given a Markov abstraction $\phi$, we can always generate another abstraction $\phi'$ by adding a procedural reshuffling of the $\phi$ representation's bits. Since the $\phi'$ representation contains all the information that was in the original representation, $\phi'$ is also a Markov abstraction. However, the new representation may be highly inefficient for learning.

To demonstrate this, we ran an experiment where we optionally relabeled the positions in the $6 \times 6$ gridworld domain, and trained two agents: one using the smooth, true $(x, y)$ positions, and one using the non-smooth, relabeled positions. Although both representations are Markov, and contain exactly the same information, we observe that the agent trained on the non-smooth positions performed significantly worse (see Figure 10).

The loss term $\mathcal{L}_{Smooth}$ used in Section 7 penalizes consecutive states that are more than $d_0$ away from each other, thereby encouraging representations to have a high degree of smoothness in addition to being Markov. This approach is similar to the temporal coherence loss proposed by Jonschkowski & Brock (2015).

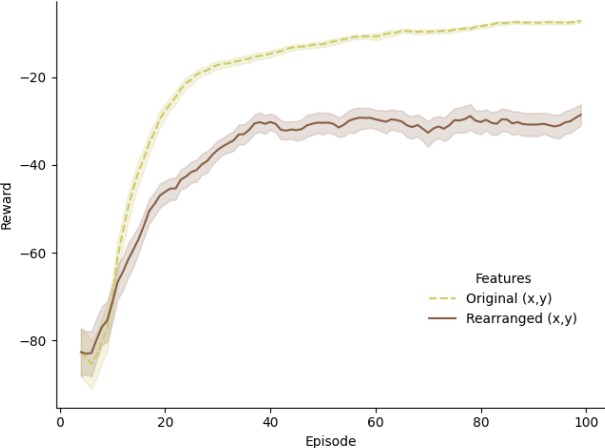

Figure 10: Mean episode reward for the $6 \times 6$ gridworld navigation task, comparing original $(x, y)$ position with rearranged $(x, y)$ position. (300 seeds; 5-point moving average; shaded regions denote 95% confidence intervals).

# M   DeepMind Control Ablation Study

We ran an ablation study to evaluate which aspects of our training objective were most beneficial for the DeepMind Control domains. We considered the RAD implementation and its Markov variant from Section 7, as well as modifications to the Markov objective that removed either the pretraining phase or the smoothness loss, $\mathcal{L}_{Smooth}$ (see Figure 11).

Overall, the ablations perform slightly worse than the original Markov objective. Both ablations still have better performance than RAD on three of six domains, but are tied or slightly worse on the others. Interestingly, removing pretraining actually results in a slight *improvement* over Markov on Finger. Removing smoothness tends to degrade performance, although, for Cheetah, it leads to the fastest initial learning phase of any method.

We suspect the results on Cheetah are worse than the RAD baseline because the experiences used to learn the representations do not cover enough of the state space. Learning then slows down as the agent starts to see more states from outside of those used to train its current representation. As we point out in Section 3.6, it can be helpful to incorporate a more sophisticated exploration strategy to overcome these sorts of state coverage issues. Our approach is agnostic to the choice of exploration algorithm, and we see this as an important direction for future work.

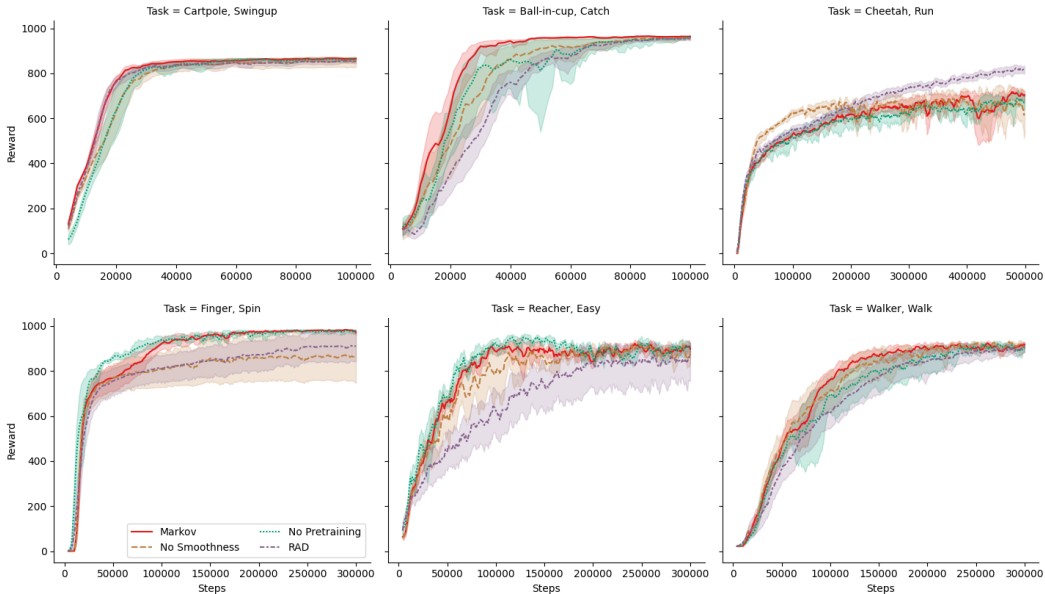

Figure 11: Ablation results for DeepMind Control Suite. Each plot shows mean episode reward vs. environment steps. (Markov and RAD – 10 seeds; others – 6 seeds; 5-point moving average; shaded regions denote 90% confidence intervals).