# OpenReview forum: "Learning Markov State Abstractions for Deep Reinforcement Learning"
_NeurIPS.cc/2021/Conference — NeurIPS 2021 Poster_

### Official Review · Reviewer_vzzj · 2021-07-15

**Rating:** 6
**Confidence:** 3

**Summary:**

# Learning Markov State Abstractions for Deep Reinforcement Learning

The paper proposes a self-supervised representation learning method for deep RL agents that ensures the learned representation is Markov and non-trivial (avoids collapse).

In a rewardless gridworld, the learned representation is competitive with an expert representation and with representations learned via inverse models. The proposed method is also shown to  improve on an RAD baseline in control tasks. Both the proposed method and the RAD baseline outperform the expert state representation. The method is SoTA in some of the tasks.

**Ethical Concerns:**

No ethical concerns.

**Limitations And Societal Impact:**

The authors have adequately addressed the limitations and potential negative societal impact of their work.

**Main Review:**

## For the response

### Major concerns:
1. How much of the improvement is due to RAD + L_inv?
2. Is the improvement to the SoTA significant?

I will improve my score if I am convinced that the answer to (1) is negative and the answer to (2) is positive and argued clearly in the text. A table with performance at 100k training steps and 500k training steps would also help put the results into perspective related to previous work, and make it easier to assess significance.

### Minor concerns:
As I argue below, the noise-contrastive estimation for the "Density Ratios" implementation can be hard to make work in other settings. Can you clarify where you expect your representation learning technique to be impactful, and where it will require further research?

## Strengths

I find the framing of the problem interesting: The paper acknowledges that if we are learning the state representation, then we are prone to losing properties that are important for solving the underlying MDP.

The proposed method also outperforms current SoTA in the DeepMind Control Suite (to the extent that I could understand from the literature) in some of the tasks at 100k steps. Breakdown: 1 improved final performance, 2 improved sample complexity with same final performance, 2 with roughly same performance throughout training, 1 with worse performance.

## Weaknesses

The empirical evaluation on control tasks makes it difficult to separate the benefits of augmentations and learning an inverse model from the benefits of the proposed method. The proposed method uses four elements: L_inv, L_ratio and L_smooth and augmentations, but are these four elements together responsible for the performance improvement? L_inv and augmentations have been used separately in previous work. Given that L_inv performs as well as the proposed method in the gridworld, it seems plausible that L_inv plus augmentations would improve on augmentations only.

## Correctness

The paper seems correct, with some gaps between the theoretical results and the method implementation.

## Clarity

The paper is clearly written, though I think some editorial changes would improve it further (see feedback). The critical point in clarity I suggest addressing is helping the reader understand how impactful the performance gains are in the DeepMind Control tasks. The gains provided by the extra losses are smaller than the gains obtained by the augmentations.

## Relation to prior work

I found the discussion informative. There's another group of papers that do not do representation learning on control tasks, but might be otherwise of interest:
* https://arxiv.org/pdf/1611.05397.pdf
* https://arxiv.org/pdf/1906.09237.pdf
* http://proceedings.mlr.press/v119/guo20g/guo20g.pdf
* https://arxiv.org/pdf/2007.05929.pdf

For the empirical evaluation, I have seen in other papers (RAD and DrQ, https://arxiv.org/pdf/2004.13649.pdf , for example) provide a table for 100k steps and 500k steps evaluations. It would benefit comparison with previous work and strengthen the SoTA claims if you could provide a similar table.

## Reproducibility

I think the text and the appendix add enough information for reproducibility.

## Additional Feedback

Thank you for submitting your paper for review. I would like to share some comments that I hope you will find useful.

The way the "Density Ratios" is implemented, by treating shuffled experience as a sample from the marginal, effectively means you are treating the marginal as stationary. It's good that this seems to work in this case, but in other cases (e.g., where experience is coming from strongly varying environments/episodes or where there's some notion of time in the state) this breaks down. In my opinion sampling from the correct marginal is one of the challenges leveraging this kind of noise-contrastive estimation across different settings. Therefore it's important to share where you expect this particular implementation of "Density Ratios" to hold up, and where further investigation is needed.

I find it interesting that the paper uses tools from POMDPs (namely belief states). My assessment is that the representation learning potentially makes the setting partially observable. Perhaps the key advantage in this setting is that one can sample from the marginal for L_ratio without a lot of work. In my experience that marginal is a lot harder to sample from in POMDPs.

Another pressing question is how much L_smooth contributes to the final result. Appendix K suggests it has a task-dependent positive or negative effect. Are you proposing this smoothness regularizer? Is it common practice to use it in DeepMind control tasks? I am divided about the presentation for this loss. On one hand, it seems to help with the SoTA performance. On the other hand, it does not quite fit with the way the claims in the paper are stated (that Markov representations lead to improved performance).

For the discussion on the representation gap, I would suggest framing that the representation learning does not really aim at learning expert state information, but rather use the richness of the environment to learn a better representation.

I would also suggest rephrasing the claims in lines 188-190 to include more of the paper's contributions, or state that the paragraph refers to the theoretical contributions.

You use the notation $x \in z$ in eq. (2) to define $B$ (see the denominator). I suggest using $x \in \phi^{-1}(z)$ instead of $x \in X : \phi(x) = z$ and of $x \in z$.

Where does $\Pi_C$ show up in Def. 2? I guess in $B$?

I could not see the definition of $y_i$ (but I could infer one).

The plot in Fig 3(b) has very small font.

In the caption for Figure 4, you state "Adding our Markov training objective leads to state-of-the-art learning performance." I suggest moving this to the main text, because it is a core claim in the paper and it needs discussion in order to be precise:
The Markov training objective includes other components (L_smooth).
Why is the method SoTA? In Cheetah run it's outperformed by RAD. In Reacher it's outperformed by CURL.


**Time Spent Reviewing:**

5

---

> ### Author Response · Authors · 2021-08-10
> **Author response: RAD + L_inv, improvement to SOTA, density ratios, smoothness, theory vs. implementation, minor clarifications**
>
> Thank you for the feedback. We will address the major concerns first, and then respond to the remaining comments below.
>
> $ $
>
> ### 1. RAD + L_inv
>
> > *How much of the improvement is due to RAD + L_inv?*
>
> > *L_inv and augmentations have been used separately in previous work. Given that L_inv performs as well as the proposed method in the gridworld, it seems plausible that L_inv plus augmentations would improve on augmentations only.*
>
> __Response:__ It is entirely possible that inverse-only would perform just as well, and as noted, we do include the inverse-only ablation in Figure 3. The difference between our approach and one that only implements the inverse model objective is that when our loss goes to zero, the Markov conditions are satisfied. By contrast, *even if* the inverse loss goes to zero, our counterexample in Section 4.2 demonstrates that this is insufficient to learn a Markov abstraction. In light of this counterexample and the increased cost of running the continuous control experiments, we felt there was little additional value to training an inverse-only model, and that our limited computational budget would be better spent comparing against published algorithms. Even if the inverse-only model performed as well as our full Markov objective on these domains, there would be nothing to suggest that it would continue to perform well on other domains, since it lacks the theoretical motivation of our combined loss.
>
> $ $
>
> ### 2. Improvement to SOTA
>
> > *Is the improvement to the SoTA significant?*
>
> > *The proposed method uses four elements: L_inv, L_ratio and L_smooth and augmentations, but are these four elements together responsible for the performance improvement?*
>
> > *The gains provided by the extra losses are smaller than the gains obtained by the augmentations.*
>
> __Response:__ Yes, the improvement to the state of the art is significant, and we will update the text to make this point clearer. In particular, our method does not require augmentations; they are an artifact of starting with RAD as a baseline. Our method is compatible with any RL algorithm, and we demonstrate its use with DQN (Section 6) and RBF-DQN (Appendix I), neither of which use augmentation. However, using some form of data augmentation (as RAD and CURL both do) has become the standard practice for these benchmarks (see also: Kostrikov et al.'s DrQ). We chose RAD as our baseline for its simplicity and the fact that it was state-of-the-art on these continuous control benchmarks. The performance improvement between our method and RAD on 4 out of 6 domains is unrelated to augmentations and is entirely due to our auxiliary objectives, since all other hyperparameters are held fixed (see "Smoothness" response below). Additionally, in most domains (Ball-in-Cup, Finger, Reacher, Walker), the improvement from RAD to Markov is at least as large as the improvement of RAD over its predecessor, CURL.
>
> -----
>
> $ $
>
> ### Density Ratios
>
> > *The way the "Density Ratios" is implemented, by treating shuffled experience as a sample from the marginal, effectively means you are treating the marginal as stationary. It's good that this seems to work in this case, but in other cases (e.g., where experience is coming from strongly varying environments/episodes or where there's some notion of time in the state) this breaks down.*
>
> __Response:__ This is correct. In the domains we consider, there is no notion of time in the state space, so any observation can occur at any timestep, and this work also assumes stationary dynamics in the ground MDP. Environments with strongly time-dependent observations do present additional challenges, although these can in principle be overcome by partitioning the observations into separate buffers by timestep, and then sampling from the appropriate buffer instead of simply shuffling the observations in the batch. However, in environments where the agent can frequently return to a state it previously visted, it's unclear that partitioning by timestep is the right thing to do, since the agent's past knowledge of a given state is likely to generalize to future timesteps in the same state.
>
> $ $
>
> ### Smoothness
>
> > *Another pressing question is how much L_smooth contributes to the final result. Appendix K suggests it has a task-dependent positive or negative effect. Are you proposing this smoothness regularizer? Is it common practice to use it in DeepMind control tasks? I am divided about the presentation for this loss. On one hand, it seems to help with the SoTA performance. On the other hand, it does not quite fit with the way the claims in the paper are stated (that Markov representations lead to improved performance).*
>
> __Response:__
>
> The Markov property is not the only important property for ensuring a good representation, but it is nevertheless a desirable one. Smoothness is another such desirable property, and its benefits in RL are well known (see: Pazis & Parr’s C-PACE algorithm; Pirotta et al.’s “Policy Gradient in Lipschitz MDPs”; Asadi et al.’s “Lipschitz Continuity in Model-Based RL”; etc.). In the text, we cite and compare against Gelada et al.’s DeepMDP, which uses Lipschitz smoothness assumptions in learning their abstract state representations, as well as Zhang et al.’s Deep Bisimulation for Control (DBC), where the representation induces a Lipschitz abstract MDP.
>
> In our experiments, smooth Markov representations appear to produce better learning performance than non-smooth ones. In Appendix J, we show this explicitly with the gridworld experiment, where performance of "Rearranged (x,y)" is roughly on-par with the Ratio model (Figure 3b) which sometimes fails to distinguish between grid cells at all. This finding is also consistent with the ablation in Appendix K, where removing the smoothness objective leads to worse performance on 5 of 6 domains (although confidence intervals overlap on Walker). We will add a discussion of smoothness to the background and related work and make it clear that the Markov property is necessary but not sufficient.
>
> $ $
>
> ### Theory vs. implementation
>
> > *The paper seems correct, with some gaps between the theoretical results and the method implementation.*
>
> __Response:__ Can you clarify what you mean by "gaps"? It is well known that neural networks trained with stochastic gradient descent are not guaranteed to converge to a global optimum, but this is not a drawback of our method---it applies to any representation learning technique that uses neural networks. Still, when our loss goes to zero, the Markov conditions are satisfied, so as neural network optimization techniques improve, our method will eventually converge to a Markov abstraction.
>
> $ $
>
> ### Minor clarifications
>
> > *In the caption for Figure 4, you state "Adding our Markov training objective leads to state-of-the-art learning performance." I suggest moving this to the main text, because it is a core claim in the paper and it needs discussion in order to be precise [...]. Why is the method SoTA? In Cheetah run it's outperformed by RAD. In Reacher it's outperformed by CURL.*
>
> __Response:__ Our intent was to communicate that in a head-to-head comparison against the previous state-of-the-art method for these DeepMind Control tasks (RAD), our method performs as well or better on a majority of domains. Our method also outperforms each of the other baselines on a majority of domains. We will update the language in the figure caption.
>
> > *Where does $\Pi_C$ show up in Def. 2? I guess in $B$?*
>
> __Response:__ Yes, we will update the expression from $B$ to $B^\pi$ to make this connection clearer.
>
> > *A table with performance at 100k training steps and 500k training steps would also help put the results into perspective related to previous work, and make it easier to assess significance.*
>
> __Response:__ We felt the learning curves were more informative, since some domains (e.g. Cartpole, Ball-in-Cup) are easy enough that the learning curves plateau long before 100K steps. In any case, we would be happy to add tables to the appendix, and we plan to release all of the learning curve data used to generate the plots when we publish the code.

---

### Official Review · Reviewer_KWGW · 2021-07-16

**Rating:** 6
**Confidence:** 4

**Summary:**

Post author response update:
Some of my concerns have been addressed and some remain. I am more positive about this paper than before the author response, so I am increasing my score.

Reasons for increasing score:
* Relationship to HOMER: The additional explanation provided by the authors (and commitment to extend explanation in the paper) reassures me about this work's novelty.
* Pre-training and performance: I am now mostly convinced that the results (with pretraining) can be compared to the baslines. Specifically, I now understand that the proposed method uses the same number of environment samples. [This is not stated in the appendix as the authors claim.] When using pretraining, their method does indeed improve upon results in past work (though final performance is generally comparable, given the 90% confidence intervals).

Remaining concerns:
* Lack of comparison to "Inverse-only": The authors choose not to compare to "Inverse-only" for Fig4 since their approach is more principled, though Inverse-only performs comparably in Fig3. The authors claim "adding our training objective to encourage Markov abstractions improves learning performance over state-of-the-art image-based RL." Since the focus is on performance, then Inverse-only performing well despite not being as principled would undermine the stated claim.
* Limited comparison of abstraction quality: The authors argue that their approach yields better abstractions and show visualizations of learned abstractions. A quantitative comparison would be more appropriate. In other [sub]fields, when one seeks to make the Markov assumption, one can measure how reasonable this assumption would be (e.g., difference between distributions when [not] conditioning on past observations). A similar approach can be taken here.

End of update


This work notes that past work on learning abstract state representations does not guarantee than the learned representation is Markovian. The authors present two conditions that, when met, ensure the representation is Markovian. Then, they introduce a method that approximately meets these two conditions. This method is compared to existing approaches where it generally learns more quickly and attains a similar final reward.

**Limitations And Societal Impact:**

Yes, limitations have been sufficiently addressed.

**Main Review:**

The motivation for learning a Markovian abstract representation is great, and the authors correctly note that past work does not guarantee that the abstract representation is Markovian while the proposed conditions would guarantee this. However, this method also does not guarantee an abstraction that is Markovian. Since this work is advocating for combining two existing objectives, the potential value is in the effectiveness of the combination. Unfortunately, the comparison to existing methods demonstrates marginal improvements over the RAD baseline (when removing the pre-training, which is provided only to the proposed method) and is missing comparisons.

The comparison to HOMER (L169-179) is crucial, and more space should be devoted in the main body to it.
Though HOMER allows only for discrete abstract states, one should compare to HOMER on an environment where continuous abstract states are better suited (domains in Figure 4?) or provide a clear reason why this cannot be done.
It is concerning that the authors note "when we convert our conditions to a training objective in Section 5, we actually recover an alternate form of the KI conditions." The proposed work is less novel than other sections (e.g., Introduction) suggest.

One of the central claims is that the proposed method is better at creating a Markovian abstract representation. An example representation is shown, but the degree to which the representation is Markovian is not measured. An introduction of some suitable metric would help demonstrate that the proposed method achieves its stated goals.

In Figure 3b, "Inverse" performs as well as Markov. The authors note this, as well as that it "does not necessarily produce Markov abstractions." However, the presented method ("Markov") does not necessarily produce Markov abstractions either, right? The degree to which the environment is Markovian is not measured. How are alpha and beta chosen to obtain an environment that is maximally Markovian?

For Figure 4, the authors note that they "pretrain a Markov abstraction on those [random] experiences for 100K steps with the reward information removed." Given that two tasks use 100K steps, 3 tasks use 300K steps, and 1 task uses 500K steps (for shown plots), using an additional 100K during pre-training favors the proposed method. For example, in Appendix H, the authors note that (for the Figure 3 experiment) providing other methods with more pre-training leads to other methods performing better and at least one method matching the proposed method's performance. The authors emphasize that the reward is not provided for these steps, but generally gathering experiences (with a specific policy) is difficult rather than obtaining rewards for certain transitions. Though these 100K steps are with a random policy, a similar pre-training step for other methods may yield better results. In Appendix K, the authors include results without pre-training for their method and acknowledge that this, in general, decreases performance (in particular, Ball-in-cup and Walker become comparable to the baseline).

Combined with the marginal improvements in the tasks where the proposed method does outperform RAD, the Figure 4 experiments do not show how the proposed method can "substantially improve learning performance over existing approaches" (Conclusion).

The introduction of a third objective is motivated well, but this change, again, raises the question of how Markovian is the learned representation (presumably, this extra objective leads to larger losses for the other two).

None of the baseline methods are equivalent to Inverse-only, correct? Given that Inverse-only performs well in Figure 3, including it in Figure 4 would show the benefits of the proposed two-objective approach. (If it is equivalent, then this should be made clear, both when discussing Figure 3 and Figure 4.)

Minor Comments:

-The list of work in L22-27 is not helpful toward understanding this work. It should be shortened / moved elsewhere.

-Figure 1 would be more helpful if it was referenced from the main text sooner (and I think 1b was not referenced at all, though it clearly shows the approach).

-The Background would benefit from a different structure (currently only 1 subsection, with remainder of comment in the "parent").

-L102-104 is out of place in the Background.

-L111-113 can be safely cut (if deemed necessary, then can be moved to end of Sec 1 or head of Sec 2 (not end of Sec 2.1)).

-In general, stating what will be shown later (e.g., "In a moment, we will define [2 lines of text] but first we must make...") is unnecessarily repetitive. Explaining everything once, in whatever order is deemed best, allows more space for other content.

-Since Section 5, Density Ratios largely follows existing work, it can be shortened to make room for other content.

**Time Spent Reviewing:**

3.5 + 2.5 (post response)

---

> ### Author Response · Authors · 2021-08-10
> **Author response: pretraining, guarantee of Markov abstraction, "Markovianity" metric, inverse-only comparison, amount of improvement, HOMER comparison, smoothness objective**
>
> Thank you for the feedback. Please see our response below:
>
> $ $
>
> ### Pretraining
>
> > *Using an additional 100K during pre-training favors the proposed method.*
>
> __Response:__ This is incorrect. We took great care to make sure the x-axis is fully comparable across all methods. As we mention in Appendix F.3, we pretrain the abstraction on exactly the same experiences that are already being used to initialize the replay buffer for RAD. All methods train for the same number of environment steps.
>
> > *For example, in Appendix H, the authors note that (for the Figure 3 experiment) providing other methods with more pre-training leads to other methods performing better and at least one method matching the proposed method's performance.*
>
> __Response:__ It should be unsurprising that the pixel prediction model eventually recovers the performance of the Markov abstraction, because the pixel prediction task is a valid way to ensure Markov abstract states. However, as we discuss in Sec. 3, the pixel prediction objective is misaligned with the fundamental goal of state abstraction, since it must effectively throw away no information. It is clear from Fig. 3 and the extended pretraining study of Appendix H that our method is able to learn a Markov representation substantially faster than pixel prediction. Again, this is unsurprising because pixel prediction is a fundamentally more challenging objective.
>
> > *For Figure 4, the authors note that they "pretrain a Markov abstraction on those [random] experiences for 100K steps with the reward information removed." [...] Though these 100K steps are with a random policy, a similar pre-training step for other methods may yield better results.*
>
> __Response:__ Again, the experiences used for pretraining are not additional experiences---they are exactly the same experiences that are already being used to initialize the replay buffer. Note that some of the baseline methods, including RAD in particular, always require reward information, so it is unclear what it would mean to apply a similar pretraining step. Essentially, pretraining RAD would be equivalent to training, and this would require additional environment steps and/or RL updates. In our experiments, our method and RAD get equal numbers of each.
>
> $ $
>
> ### Guarantee of Markov abstraction
>
> > *This method [...] does not guarantee an abstraction that is Markovian.*
>
> > *In Figure 3b, "Inverse" performs as well as Markov. The authors note this, as well as that it "does not necessarily produce Markov abstractions." However, the presented method ("Markov") does not necessarily produce Markov abstractions either, right?*
>
> __Response:__ It is well known that neural networks trained with stochastic gradient descent are not guaranteed to converge to a global optimum, but this is not a drawback of our method---it applies to any representation learning technique that uses neural networks. The difference between our approach and one that only implements the inverse model objective is that when our loss goes to zero, the Markov conditions are satisfied. By contrast, *even if* the inverse loss goes to zero, our counterexample in Sec. 4.2 demonstrates that this is insufficient to learn a Markov abstraction. As neural network optimization techniques improve, our method will converge to a Markov abstraction; the inverse-model-only approach will not.
>
> $ $
>
> ### "Markovianity" metric
>
> > *One of the central claims is that the proposed method is better at creating a Markovian abstract representation. An example representation is shown, but the degree to which the representation is Markovian is not measured. An introduction of some suitable metric would help demonstrate that the proposed method achieves its stated goals.*
>
> __Response:__
>
> A state space is either Markov, or it is not, by Def. 1. The notion of a "Markovianity" metric isn't something that is well defined, and we are unaware of prior work in this area. One could in theory use our Markov loss as a measure of “how Markov” the representation is, with lower loss values corresponding to “more Markov” representations, although it’s unclear how to calibrate that, since the difficulty of learning an inverse model or a discriminator may vary depending on the domain.
>
> The example abstract representation in Fig. 3a is Markov. It shows clearly defined clusters of abstract states in a 6x6 grid pattern matching that of the underlying Markov state space used to generate the visual observations. Not only does our method reliably learn a Markov state space, it does so more quickly than other approaches, especially when contrasted with the representation produced by L_ratio alone or the pixel reconstruction/prediction results in Appendix G, some of which failed to differentiate between grid positions at all.
>
> For the continuous control domains, the abstract state space is 50-dimensional and difficult to visualize. This is precisely why it's so important to connect the training objective to the desired theoretical property we wish to incentivize, rather than relying on metrics that approximate it or on qualitative visual assessments.
>
> > *How are alpha and beta chosen to obtain an environment that is maximally Markovian?*
>
> __Response:__ Rather than thinking of alpha and beta as ensuring a maximally Markovian representation, it's better to think of them as compensating for the relative difficulty of learning an inverse model and a discriminator for the domain in question. In practice, they are tuned over a range of values, and we document the ranges we considered in Appendix F.3.
>
> $ $
>
> ### Inverse-only comparison
>
> > *None of the baseline methods are equivalent to Inverse-only, correct? Given that Inverse-only performs well in Figure 3, including it in Figure 4 would show the benefits of the proposed two-objective approach.*
>
> __Response:__ As you mentioned, we do include the inverse-only ablation in Fig. 3. However, in light of the counterexample in Sec. 4.2 and the increased cost of running the continuous control experiments, we felt there was little additional value to training an inverse-only model, and that our limited computational budget would be better spent comparing against published algorithms. Even if the inverse-only model performed as well as our full Markov objective on these domains, there would be nothing to suggest that it would continue to perform well on other domains, since it lacks the theoretical motivation of our combined loss.
>
> $ $
>
> ### Amount of improvement
>
> > *Unfortunately, the comparison to existing methods demonstrates marginal improvements over the RAD baseline*
>
> > *Combined with the marginal improvements in the tasks where the proposed method does outperform RAD, the Figure 4 experiments do not show how the proposed method can "substantially improve learning performance over existing approaches"*
>
> __Response:__ This seems to be a disagreement about what constitutes a substantial improvement in learning performance. Compared to RAD, our method learns faster on 4 domains, equally fast on 1, and slower on 1, achieving roughly the same final performance (better in 1 domain, worse in 1, equal in 4). An even more favorable comparison can be made with any of the other baselines, which means our method is a state-of-the-art technique. Our approach even represents a marginal improvement over a hypothetical "best of" oracle which always chooses the best performing baseline. Additionally, in most domains (Ball-in-Cup, Finger, Reacher, Walker), the improvement from RAD to Markov is at least as large as the improvement of RAD over its predecessor, CURL.
>
> $ $
>
> ### HOMER comparison
>
> > *The comparison to HOMER [...] is crucial, and more space should be devoted in the main body to it. Though HOMER allows only for discrete abstract states, one should compare to HOMER on an environment where continuous abstract states are better suited [...] or provide a clear reason why this cannot be done. It is concerning that the authors note "when we convert our conditions to a training objective in Section 5, we actually recover an alternate form of the KI conditions." The proposed work is less novel than other sections [...] suggest.*
>
> __Response:__ The HOMER work is indeed related, but distinct, and we would be happy to add more discussion to the main text. First, the (novel) theoretical formulation of Markov abstractions is completely absent from the HOMER paper. Second, as noted, HOMER was designed for discrete abstract states and has not been demonstrated to work for continuous state spaces, whereas ours works for both. HOMER additionally requires specifying---in advance---an upper bound on the *number* of abstract states (N), which is clearly impossible for continuous state spaces. For a hypothetical continuous adaptation of HOMER, even if one were to provide an N that was sufficiently large to practically cover the continuous state space at some fine-grained length scale, the algorithm would then need to find a policy cover to reach each of those individual abstract states, which is impractical. Moreover, there is no obvious way to disentangle HOMER's representation learning algorithm from its exploration scheme, so it would be challenging to ensure a meaningful comparison.
>
> $ $
>
> ### Smoothness objective
>
> > *The introduction of a third objective is motivated well, but this change, again, raises the question of how Markovian is the learned representation (presumably, this extra objective leads to larger losses for the other two).*
>
> __Response:__ Introducing a third objective to incentivize smoothness does not necessarily imply larger losses for the other two objectives. Our example in Appendix J demonstrates that one Markov representation can be smoother than another while containing exactly the same information w.r.t. the Markov property, and that the smoother representation leads to faster learning. Our smoothness objective simply nudges the neural network towards this more useful Markov representation.

---

### Official Review · Reviewer_tW7x · 2021-07-16

**Rating:** 7
**Confidence:** 3

**Summary:**

This paper discusses conditions which are sufficient for a state representation to maintain the Markov property, and describes how a pair of practical objectives can encourage these conditions to be satisfied. An empirical investigation shows these auxiliary objectives lead to structured representations and good sample efficiency.

**Limitations And Societal Impact:**

In the main review I've made suggestions for some areas of potential limitations that could be discussed further.

**Main Review:**

This is an interesting paper that connects some previously-studied representation-learning approaches with the principle of preserving the Markov property in an abstract MDP.
Overall the paper is fairly well written, and the contributions are placed well in the context of prior work.

Nonetheless, some aspects of the method could deserve further justification or investigation, and I have a few questions for clarifications:

I think further effort should be made to justify the Markov property as an appropriate driving property for representation learning. Certainly many RL algorithms assume a Markov state, and it’s fairly clear why this would be important to preserve when using a representation learning phase followed by learning with that fixed representation.
It’s less clear why this property needs to be preserved for some specific part of the representation (e.g. penultimate activations) if the policy conditions on a Markov state or on a Markovian history, as in normal ‘end-to-end’ deep RL.
This point could be argued more strongly, and any empirical evidence to suggest that deficiencies of other representation learning objectives are specifically due to a lack of Markovianity would be valuable!

I’m also curious in what settings the authors think preserving the Markov property is too strong. One obvious situation is related to the generic deficiency of reward-agnostic representation learning; if there are distractor features and dynamics that are irrelevant to any reward functions of interest they must still be captured in this case. Of course, there are other advantages to being agnostic to reward, but this should probably be discussed.
Another potential point of comparison is, as far as I understand it, between bisimulation (cares about all Pi), the Markov-preserving abstraction (cares about Pi_phi), and *pi-bisimulation* (cares about pi only): is the latter not strong enough a condition? Do the authors think that the coarsest Markovian state abstraction would be in some way optimal?

A clarification: when the authors write that the method can capitalise on reward when available, does this just refer to the fact that the representation learning objectives can be used as an auxiliary alongside normal reward-maximising RL? Or have I missed some other way reward information is incorporated?

Regarding the experiments, can the authors elaborate on the initialisation of the replay buffer and the pretraining phase used? (i.e. is the x-axis in fig 4 fully comparable between all methods?).
The results are nice, if not astounding, which is unsurprising as the method builds on previously-explored objectives.

Another point of feedback: I found the figures difficult to read, with many overlapping thin lines. Maybe the learning curves could be summarised as bar charts showing the area under curve, with full curves in the appendix at larger size?


**Time Spent Reviewing:**

4

---

> ### Author Response · Authors · 2021-08-10
> **Author response: Usefulness of Markov property, distractor features, full- and $\pi$-bisimulation, reward information, initialization and pretraining, presentation of results**
>
> Thank you for your feedback. Please see our response below:
>
> $ $
>
> ### Usefulness of Markov property
>
> > *I think further effort should be made to justify the Markov property as an appropriate driving property for representation learning. Certainly many RL algorithms assume a Markov state, and it’s fairly clear why this would be important to preserve when using a representation learning phase followed by learning with that fixed representation. It’s less clear why this property needs to be preserved for some specific part of the representation (e.g. penultimate activations) if the policy conditions on a Markov state or on a Markovian history, as in normal ‘end-to-end’ deep RL. This point could be argued more strongly, and any empirical evidence to suggest that deficiencies of other representation learning objectives are specifically due to a lack of Markovianity would be valuable!*
>
> __Response:__
>
> While in principle one could simply train an agent end-to-end using the raw (Markov) images as input, in practice RL often performs worse without additional representation learning objectives, as evidenced by the large number of different methods designed for that very purpose.
>
> As you say, the Markov property is particularly important for pretraining representations without reward and then learning with the fixed representation. However, it can also be useful for *initializing* a representation that will be fine-tuned during the course of RL (see results for Ball-in-Cup, where the initial learning curve is much steeper than other baselines), and/or as a concurrent objective during the entire learning process (see results for Finger, where performance escapes the plateau that both RAD and CURL fall into).
>
> Here we focus on the single-task setting, but we expect Markov abstractions will also be useful in the multi-task setting, since they can be learned reward-free and before any task rewards are observed. Furthermore, if we want to use the representations for model-based RL, the Markov property is necessary for learning Markov transition and reward models.
>
> $ $
>
> ### Distractor features
>
> > *I’m also curious in what settings the authors think preserving the Markov property is too strong. One obvious situation is related to the generic deficiency of reward-agnostic representation learning; if there are distractor features and dynamics that are irrelevant to any reward functions of interest they must still be captured in this case. Of course, there are other advantages to being agnostic to reward, but this should probably be discussed.*
>
> __Response:__
>
> Distractor features are a good example to discuss. Of course, if we know which features are distractors, the best abstraction is one that throws that information away. Reward information can help identify which features are salient and which are distractors. Additionally, if we know the optimal policy, we can abstract even further, such that we only retain a single abstract state for each unique action selected by the policy (e.g. $\pi^*$-irrelevance abstraction).
>
> However, in the absence of such information, it is impossible to know which features are distractors and which are not, and the best we can do is to simply retain everything that behaves predictably. The same features that are distractors in one task might actually be important for another task, and without rewards, it is impossible to distinguish these. Still, if there are distractor features which do *not* behave predictably, such as the per-pixel noise we inject into the visual gridworld images, our method has no problem ignoring them.
>
> $ $
>
> ### Bisimulation and $\pi$-bisimulation
>
> > *Another potential point of comparison is, as far as I understand it, between bisimulation (cares about all Pi), the Markov-preserving abstraction (cares about Pi_phi), and pi-bisimulation (cares about pi only): is the latter not strong enough a condition?*
>
> __Response:__ One way to view Markov abstractions is that they are a useful compromise between full bisimulation and $\pi$-bisimulation. The former can be too strong, since it constrains the representation based on policies the agent may never actually select. The latter can be too weak, because it only tells whether two states are behaviorally equivalent *under the policy $\pi$*. If the policy deviates from $\pi$, e.g. during learning, the metric must be updated (and the representation along with it).
>
> > *Do the authors think that the coarsest Markovian state abstraction would be in some way optimal?*
>
> __Response:__ It's important to distinguish between a Markov state representation (Def. 1) and a Markov abstraction (Def. 2). A trivial abstraction that collapses every ground state to a single abstract state would still result in a Markov state representation (Def. 1). However, the coarsest Markov abstraction (Def. 2) is one which leads to the smallest possible abstract state representation that does not introduce partial observability.
>
> $ $
>
> ### Clarification about reward information
>
> > *A clarification: when the authors write that the method can capitalise on reward when available, does this just refer to the fact that the representation learning objectives can be used as an auxiliary alongside normal reward-maximising RL? Or have I missed some other way reward information is incorporated?*
>
> __Response:__ Yes, reward-maximizing RL is the traditional way to incorporate reward information, although one could also use the abstract state to estimate an explicit reward model.
>
> $ $
>
> ### Initialization and pretraining
>
> > *Regarding the experiments, can the authors elaborate on the initialisation of the replay buffer and the pretraining phase used? (i.e. is the x-axis in fig 4 fully comparable between all methods?).*
>
> __Response:__
>
> Yes, we took great care to make sure the x-axis is fully comparable across all methods. As we mention in Appendix F.3, we pretrain the abstraction on exactly the same experiences that are already being used to initialize the replay buffer. All methods train for the same number of environment steps.
>
> By default, RAD uses 1K `init_steps` to initialize the replay buffer for all domains, which is only 2 episodes on domains with `action_repeat` 2. To ensure adequate coverage of the state-action space to train the representation, we increased this initialization period to equal 10 episodes worth of experience across all domains. After generating these 10 episodes of initialization experiences, we use them to pretrain the abstraction for 100K gradient updates. (These pretraining updates do not use any additional environment experience, they simply reuse the experiences that were already used to initialize the replay buffer.)
>
> After our method finishes pretraining, RAD will have already performed (`init_steps` - 1K) RL updates, so we compensate for our method’s longer initialization period by adding (`init_steps` - 1K) catchup learning steps. This again uses the experiences that were already in the replay buffer and simply ensures that RAD and our method perform the same number of RL updates. Note that our method is still at a slight disadvantage for smaller `action_repeat` settings, since RAD can begin changing its policy based on reward information after just 1K steps, but our method must wait as much as 5K steps for some domains.
>
> $ $
>
> ### Presentation of results
>
> > *Another point of feedback: I found the figures difficult to read, with many overlapping thin lines. Maybe the learning curves could be summarised as bar charts showing the area under curve, with full curves in the appendix at larger size?*
>
> __Response:__ Thank you for this feedback. We will improve the presentation of these results for the final version.

---

### Official Review · Reviewer_br5G · 2021-07-18

**Rating:** 6
**Confidence:** 3

**Summary:**

Summary
-------


This paper analyzes the problem of learning Markov state abstractions in
reinforcement learning. The authors first prove that the abstraction
must satisfy two conditions: an inverse model equality and a density
ratio equality. These two conditions cannot be enforced in the training
objective, so an approximate regularizer is proposed instead. The
proposed algorithm is compared with several baselines in an offline
learning regime (visual gridworld) which evaluates the quality of the
state abstraction and an online learning regime (continuous control)
which evaluates the performance.



**Limitations And Societal Impact:**

The authors adequately addressed the limitations and potential negative societal impact of their work.

**Main Review:**

Decision
--------

Overall, the paper provides a well-written analysis of learning Markov
state abstractions and is a valuable addition to the RL literature. The
paper is held-back by the lack of insight in the experimental
investigations. While the experiments on GridWorld show that their
proposed method does learn a Markov state abstraction, their further
experiments do not clearly show the benefit of a Markov state
abstraction. Taken together, I am rating this paper tentatively at a 5
("Marginally below the acceptance threshold").


Strengths
---------

-   The theoretical analysis of learning Markov abstractions is
    interesting and novel. I particularly like that the work highlights
    that the density ratio condition is often ignored in the previous
    literature, comments on why this might be the case and later shows
    it through an ablation study in Section 6. This "closing the loop"
    from theory to empirics is excellent.
-   Presentation of the work and the writing quality is very high. While
    some aspects of the technical sections are difficult to follow, I
    think the paper does a superb job of motivating and explaining the
    decisions made throughout the paper.

Weaknesses
----------


-   The experiments are not well motivated, and do not meaningfully
    answer whether learning Markov abstractions is a worthwhile
    endeavor.
-   While out of scope for this paper, I think there is a lacking
    discussion on the trajectory/memory component of learning Markov
    state representations. Of course, the goal of this paper is to study
    state abstractions. However, the Markov property is less important
    for high dimensional observations like images (where, often the
    images are used as-is). The real interest and challenge lies in
    learning a Markov state by remembering information from previous
    observations.

Detailed Comments
-----------------
-   Section 4 builds on technical machinery from Section 2 in such a way
    that I often needed to go back and forth between the two sections. I
    feel that part of the problem is that the two sections are separated
    by related work. However, I do not believe that switching section 2
    and 3 is necessarily the solution. Instead, I think the authors
    should contextualize some of the technical machinery such as the
    belief distributions using their working example (the MDP in figure
    2).

-   Section 7: I think the results here, while impressive, completely
    undermine the rest of the paper. If Markov states are important,
    then surely "expert" Markov states are the gold standard for state
    representation. And yet, you show that this is often not the case.
    Indeed, your method and RAD seem to outperform the expert in many
    cases. If this is true, then what is the motivation for learning a
    markov abstraction to begin with? And what is the reason for the
    proposed algorithm's success?

Minor Comments
--------------

-   Figure 3b: The experiment is in the offline learning section, yet
    the x-axis shows performance over a number of episodes. This seems
    to suggest that the experiment is actually online.
-   Figure 3b, 4: Y-axis says reward, should be return?
-   Line 290: How exactly is it mapped to an image? Are the pixels
    equidistant and what is the resolution for the image?

Post Rebuttal
--------------
In light of the discussion below, I have increased my score from a 5 -> 6.

**Time Spent Reviewing:**

6

---

> ### Author Response · Authors · 2021-08-10
> **Author response: expert representations, contextualizing technical machinery, experiment clarification, abstractions over histories**
>
> Thank you for your comments. Please see the response below:
>
> $ $
>
> ### "Expert" Representations
>
> > *Section 7: I think the results here, while impressive, completely undermine the rest of the paper. If Markov states are important, then surely "expert" Markov states are the gold standard for state representation. And yet, you show that this is often not the case. Indeed, your method and RAD seem to outperform the expert in many cases. If this is true, then what is the motivation for learning a markov abstraction to begin with? And what is the reason for the proposed algorithm's success?*
>
> __Response:__
>
> Not all "expert" representations are created equal. Even though the "SAC (expert)" agent exhibits poor performance relative to some of the visual methods, that does not undermine the central claim of the paper, since other alternative Markov representations exist that may be even more beneficial for RL. If all Markov representations were of equal quality, simply running SAC on frame-stacked image inputs ought to be enough, and none of these methods would need to do representation learning at all.
>
> As we show in Appendix J, simply relabeling Markov states to produce a new (non-smooth, but still Markov) representation is enough to significantly degrade performance. Additionally, in some of our informal gridworld experiments (not included in the submission, but we'd be happy to add them to the appendix), we observed that a one-hot expert representation, indicating which position in the grid was occupied (also Markov), substantially improved on the original "expert" (x,y) representation.
>
> As we mention in the introduction, the Markov property is only one of the many potentially desirable properties that we might want our representations to have. The explanation for why Expert SAC is worse than some visual methods (including ours) is likely that its so-called "expert" representation may in fact be missing other desirable properties unrelated to the Markov property. However, this doesn't make the Markov property any less important, and indeed, our results show that adding the Markov objective to an existing state-of-the-art visual method (RAD) leads to learning performance that is significantly better on 4 domains, worse on 1, and tied on 1.
>
> $ $
>
> ### Contextualizing technical machinery
>
> > *Section 4 builds on technical machinery from Section 2 in such a way that I often needed to go back and forth between the two sections. I feel that part of the problem is that the two sections are separated by related work. However, I do not believe that switching section 2 and 3 is necessarily the solution. Instead, I think the authors should contextualize some of the technical machinery such as the belief distributions using their working example (the MDP in figure 2).*
>
> __Response:__ We tried to present enough material in the background section to allow for a meaningful discussion of the related work. We appreciate the suggestion to extend the working example, using Fig. 2 to further contextualize the belief distributions, and we are happy to do so. Please let us know if there are other specific details that can be moved between Sections 2 and 4 that would make the material more accessible.
>
> $ $
>
> ### Experiment clarification
>
> > *Figure 3b: The experiment is in the offline learning section, yet the x-axis shows performance over a number of episodes. This seems to suggest that the experiment is actually online.*
>
> __Response:__ The experiment in section 6 is about offline learning *of the abstraction*. We freeze the abstraction after pretraining and subsequently use it for traditional (online) RL. The plot shows learning performance during this (online) RL period, with episodes on the x-axis.
>
> > *Figure 3b, 4: Y-axis says reward, should be return?*
>
> __Response:__ The y-axis is reward-per-episode. Plotting return-per-episode would also be appropriate, since we do still use a discount factor (gamma=0.9) with DQN, but because the step reward is -1 everywhere and there is a single terminal state, the agent will have the same preference rankings over trajectories/policies whether we use return or reward. The original DQN paper used reward rather than return, and most deep RL papers now do the same.
>
> > *Line 290: How exactly is [the gridworld position] mapped to an image? Are the pixels equidistant and what is the resolution for the image?*
>
> __Response:__ We describe the process for mapping (x,y) positions to images in Appendix E. The image displays each position in the 6x6 grid as a 3px x 3px patch, inside of which we light up one pixel (in the center) and then smooth it using a truncated Gaussian kernel. This results in an 18x18 image (where 3px x 3px grid cells are equidistant), to which we then add per-pixel noise from another truncated Gaussian.
>
> $ $
>
> ### Abstractions over histories
>
> > *While out of scope for this paper, I think there is a lacking discussion on the trajectory/memory component of learning Markov state representations. Of course, the goal of this paper is to study state abstractions. However, the Markov property is less important for high dimensional observations like images (where, often the images are used as-is). The real interest and challenge lies in learning a Markov state by remembering information from previous observations.*
>
> __Response:__ While it's true that images are often used "as-is", we do see a clear benefit in both experiments from using abstract representations that explicitly minimize our Markov objective. In any case, we agree that learning abstractions over histories is certainly an interesting direction for future work. One paper which explores that direction from a theoretical perspective, and which this work draws inspiration from, is Marcus Hutter's "Extreme State Aggregation beyond MDPs". In that paper, Hutter discusses approximate aggregation of histories using a "dispersion probability" $B(h|z,a)$, similar to the belief distribution we use, where $h := \\{x_0, x_1, \\cdots\\}$ is a sequence of observed states. Hutter proves several useful theoretical properties using this formulation and provides a rough sketch of what a hypothetical representation learning algorithm based on that formulation might look like. We would be happy to add this additional discussion to the paper.

---

### Decision · Program_Chairs · 2021-09-27

**Decision:**

Accept (Poster)

**Comment:**

All reviewers agree that the proposed approach is an important contribution to the area of representation learning in reinforcement learning. Although some lingering concerns regarding the empirical evaluation of the method still remain, overall the reviewers think that the paper introduces a fairly elegant and potentially useful recipe for enforcing the Markov property.

We strongly advise the authors to carefully consider the suggestions made in the reviews and also during the discussion phase. For example, the fact that there exist multiple state abstractions that satisfy the Markov property and that other, additional, properties may also play an important role is a point that should be more clear in the paper. Another observation that came up multiple times in the reviews and was a point of contention in the discussion that followed was the lack of comparisons with a version of the proposed approach that only uses the “inverse loss”. The authors should add a comment to the paper explaining the reasons not to have such a comparison. There were also a few concerns regarding the presentation that should be taken into account when preparing a new version of the submission.